# SOURCE-FREE TEST-TIME ADAPTATION FOR DIFFUSION-BASED VIRTUAL TRY-ON

## ABSTRACT

The rapid growth of e-commerce has driven notable advancements in diffusion-based virtual try-on models. Virtual try-on models, however, suffer significant quality degradation when deployed on real-world data that differs from their source (training) distribution. To address challenges in quality degradation due to domain shifts, we introduce a test-time adaptation framework that enhances try-on quality during diffusion denoising (inference time) without requiring model retraining or updates to the original network parameters. We introduce statistical distribution matching across complementary domains during the diffusion denoising process. Comprehensive evaluation across four state-of-the-art diffusion models (IDM-VTON, LaDI-VTON, Stable-VTON, TPD) and three datasets (VITON-HD, DressCode, DeepFashion) demonstrates notable improvements across multiple dataset-method combinations, with sharpness gains averaging 7.74% and distortion reduction of 0.95%. Our approach addresses important practical challenges in commercial virtual try-on deployment, enabling quality improvements across diverse domain conditions while preserving the original model's capabilities.

## 1 INTRODUCTION

Virtual try-on is a key application in e-commerce, enabling users to digitally wear garments or visualize them on template models. Over the years, advancements in try-on technology have focused on efficiently "warping" garments onto a target person while preserving intricate details. Early machine learning-based approaches Han et al. (2018); He et al. (2022); Wang et al. (2018) achieved some success in detail retention but were limited by low resolution and common artifacts such as blurring, color bleeding, and texture inconsistencies.

Appearance flow-based methods Zhou et al. (2016); Shim et al. (2024); Gou et al. (2023); Yu et al. (2023); Chen et al. (2023b) have treated virtual try-on as a view synthesis problem, while 3D geometric formulations Zhao et al. (2021) attempt to map input image pixels to their expected positions in the warped garment. However, these methods often struggle with generalization, as models trained on one dataset frequently fail when applied to unseen data. Hybrid approaches, such as DCI-VITON Gou et al. (2023), integrate appearance flow with diffusion models, yet they still exhibit distortions when handling new garment types or poses.

Denoising diffusion probabilistic models (DDPMs) have shown strong performance in image-to-image tasks such as segmentation, object detection, and virtual try-on Tian et al. (2023); Chen et al. (2023a); Gou et al. (2023); Kim et al. (2024); Zhu et al. (2023). These models are typically conditioned on input images and garments, sometimes using text for finer control Gou et al. (2023). While they preserve texture well Yang et al. (2024) and generalize across domains, they struggle with unseen garment styles and poses due to their stochastic nature. To address domain shifts in virtual try-on, we propose the first test-time adaptation (TTA) framework for diffusion models that refines generation during inference without modifying model parameters. Unlike prior methods that require retraining Xing et al. (2024), our approach is source-free and adapts per image. While TTA has seen use in classification Wang et al. (2021); Boudiaf et al. (2022); Niu et al. (2022); Osowiechi et al. (2024), its potential for image-to-image tasks remains largely unexplored.

Our method refines denoised outputs across multiple diffusion steps using a novel multi-domain statistical loss comprising: (1) Image Domain Consistency – Aligning pixel intensity distributions for accurate color and illumination, (2) Frequency Domain Consistency – Preserving garment details

using Discrete Cosine Transform (DCT), and (3) Local Structural Consistency – Maintaining spatial coherence through local distribution matching. Our approach is 'training-free' as it does not modify the parameters ($\Theta$) of the pretrained diffusion model. It operates as a plug-and-play guidance mechanism during inference, preserving the original model's integrity while adapting its output to new data. Our key contributions are: (1) A novel guidance mechanism that adapts diffusion denoising through statistical distribution matching without modifying network parameters or requiring source training data, (2) Three complementary loss functions operating across image, frequency, and local structure domains, with theoretical convergence guarantees for the optimization process, (3) Comprehensive evaluation demonstrating consistent improvements across four state-of-the-art models and three datasets, with notable gains in semantic consistency (up to 4.2% CLIP improvement) and visual sharpness (up to 56.8% enhancement).

## 2 RELATED WORK

### 2.1 VIRTUAL TRY-ON

Virtual try-on has evolved significantly from early coarse-to-fine pipelines such as VITON Han et al. (2018), which combined warping and refinement modules to mitigate blurry artifacts. CP-VTON Wang et al. (2018) introduced geometric matching and thin-plate spline transformations for better garment alignment, while style-based frameworks like StyleGAN-VITON He et al. (2022) incorporated local context to improve realism and texture consistency. Subsequent work addressed scalability and resolution challenges. Outfit-VITON Neuberger et al. (2020) enabled multi-garment try-on with compositional architecture, while VITON-HD Choi et al. (2021) and HR-VITON Lee et al. (2022) pushed towards high-resolution synthesis ($1024 \times 768$), employing segmentation-guided metrics and joint learning of warping and parsing for improved realism in occluded regions. However, many of these methods remain sensitive to domain shifts in pose, lighting, and garment styles, struggling with generalization when applied to unseen data distributions.

Figure 1: TTA enhances image quality by preserving garment details (top), patterns (middle), and object shapes/positions (bottom).

### 2.2 DIFFUSION MODELS FOR VIRTUAL TRY-ON

Denoising Diffusion Probabilistic Models (DDPMs) Ho et al. (2020) have recently emerged as powerful generative frameworks, achieving high-fidelity generation under complex conditions. Stable Diffusion Rombach et al. (2022) demonstrated remarkable generalization capabilities via text conditioning and fine-tuning approaches. In virtual try-on, diffusion-based models such as DCI-VTON Gou et al. (2023), LaDI-VTON Morelli et al. (2023), TPD Yang et al. (2024), and TryOnDiff Zhu et al. (2023) produce high-resolution results using both paired and unpaired training data. While they preserve texture well Yang et al. (2024) and show improved generalization compared to earlier methods, they often fail to preserve fine-grained details such as logos, textures, and garment boundaries under domain shifts, frequently introducing hallucinated artifacts beyond the clothing region. CAT-DM Zeng et al. (2024) integrates ControlNet for enhanced conditioning but relies on parameter updates during inference, increasing computational cost and requiring access to source training data.

### 2.3 TEST-TIME ADAPTATION

Test-time adaptation aims to bridge domain gaps by adapting models using unlabeled test data without access to source training data. Entropy minimization methods such as TENT Wang et al. (2021) update batch normalization statistics to align distributions, while parameter-

free methods like LAME Boudiaf et al. (2022) and EATA Niu et al. (2022) preserve source model integrity by filtering uncertain samples and avoiding parameter modifications. Recent works explore single-image adaptation scenarios. SITA Khurana et al. (2021) applies entropy-based consistency across class-preserving augmentations, while S-ITTA Janouskova et al. adapts segmentation masks in zero-shot settings. However, these approaches are fundamentally designed for classification and segmentation tasks, relying on discrete predictions, categorical entropy, and batch normalization statistics that are absent in generative diffusion models. However, these approaches are fundamentally incompatible with generative diffusion models due to key architectural and objective mismatches: (1) classification TTA relies on discrete class predictions and categorical entropy, while diffusion operates in continuous latent spaces; (2) existing methods depend on batch normalization updates, but diffusion models use layer/group normalization; (3) classification TTA leverages prediction confidence as pseudo-supervision, which is unavailable in image generation tasks. To our knowledge, this is the first TTA method

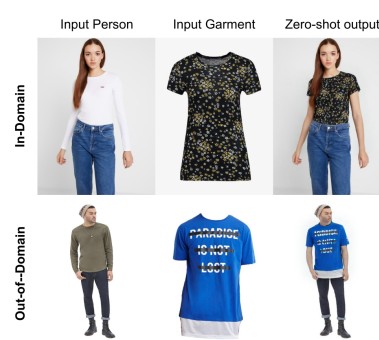

Figure 2: Illustration of domain shift in virtual try-on: Aligned domains (top) yield clear garment details, while shifts in lighting or style (bottom) cause blurred textures and distorted patterns.

for diffusion-based image-to-image generation. We propose a source-free framework using statistical distribution matching during denoising, avoiding the discrete prediction dependencies of traditional TTA methods.

## 3 PROPOSED METHODOLOGY

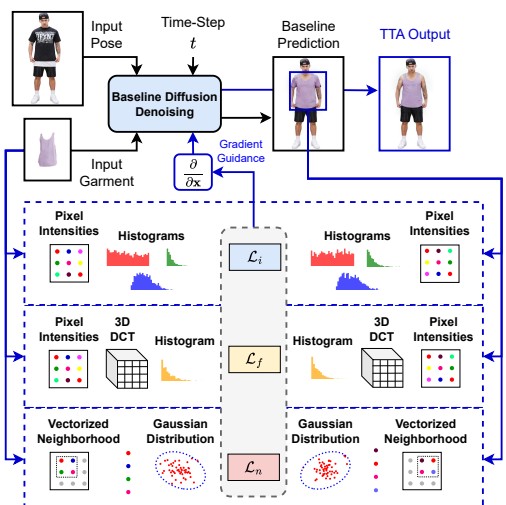

Figure 3: Overview of the proposed TTA framework. Blue arrows denote TTA guidance, black arrows denote zero-shot prediction. Guidance losses $\mathcal{L}_i$, $\mathcal{L}_f$, and $\mathcal{L}_n$ refine predictions during denoising, enabling plug-and-play TTA.

We propose a source-free TTA framework to improve virtual try-on quality (Fig. 3), addressing domain shift failures such as illegible text and distorted stripe patterns (Fig. 2). Domain shifts hinder e-commerce deployment and user experience. Our method mitigates this by guiding diffusion with statistical matching of color, texture, and structure to preserve garment fidelity across domains.

### 3.1 GUIDED DENOISING DIFFUSION

DDPMs Ho et al. (2020) are a recent class of generative models that have shown tremendous success in tasks such as conditional generation, inpainting, and translation. The ability of DDPMs to generalize well stems from learning a first-order Markov process from Gaussian noise to structured outputs. Given an input image $\mathbf{x}_0$, noise is progressively added through a fixed Markov process for $T$ time-steps until $\mathbf{x}_T$ is distributed as a standard Gaussian $\mathcal{N}(\mathbf{0}, \mathbf{I})$. In other words, a sequence of data $\{\mathbf{x}_i\}$, $i = 0, 1, \cdots, T$ is sampled with Gaussian noise with variance $\beta_t \in (0, 1)$ at time-step $t$ during the forward process as

$$q(\mathbf{x}_t|\mathbf{x}_0) = \sqrt{\bar{\alpha}_t}\mathbf{x}_0 + \sqrt{1 - \bar{\alpha}_t}\epsilon \tag{1}$$

where $\epsilon \sim \mathcal{N}(\mathbf{0}, \mathbf{I})$ is the noise added to the input, $\bar{\alpha}_t = \prod_{j=1}^{t} \alpha_j$, and $\alpha_t = 1 - \beta_t$ for $t = 1, 2, \ldots, T$ is the noise scheduling parameter. The distributions of the forward process are denoted

by $q(\mathbf{x}_t|\mathbf{x}_{t-1})$, which are assumed to follow a first-order Markov process. The reverse (or denoising process) is modeled using a parametric family of distributions denoted as $p_\theta(\mathbf{x}_{t-1}|\mathbf{x}_t)$.

The idea is that the distribution of $\mathbf{x}_T$ can be approximated reasonably well by the standard Gaussian distribution i.e., $q(\mathbf{x}_T) \sim \mathcal{N}(\mathbf{0}, \mathbf{I})$. The goal of the DDPM is to estimate the parameters of $p_\theta$ by optimizing a variational lower bound on the data likelihood $\mathbf{x}_0$ under the model $p_\theta$. The loss function optimized by the DDPM at a time-step $t$ is as follows:

$$L_t = \mathop{\mathbb{E}}_{\mathbf{x}_0,\epsilon}\Big[\|\epsilon - \epsilon_{\boldsymbol{\theta}}(\mathbf{x}_t,t)\|^2\Big], \tag{2}$$

where $\epsilon$ and $\epsilon_\theta(\mathbf{x}_t, t)$ correspond to the input (real) noise and the predicted noise at time step $t$, respectively. The denoised image at any time step $t$ is given by

$$\mathbf{x}_{t-1} = \frac{1}{\sqrt{\alpha_t}}\left(\mathbf{x}_t - \frac{1-\alpha_t}{\sqrt{1-\bar{\alpha}_t}}\epsilon_{\boldsymbol{\theta}}(\mathbf{x}_t,t)\right) + \sigma_t\epsilon, \tag{3}$$

where $\sigma_t$ is the variance of the noise and the goal is to bring the distribution of $\mathbf{x}_0$ as close as possible to that of source domain data. However, the aforementioned relation makes the reverse process slow due to the requirement of all past denoising outputs to compute a single output. To speed up the reverse process, DDIM Song et al. (2020) sampling was introduced, where multiple time-steps can be skipped and the denoised outputs can still be computed in an efficient manner. To this end, the DDIM sampling process is described as

$$\mathbf{x}_{t-1}^g = \sqrt{\bar{\alpha}_{t-1}}(\mathbf{x}_t^g - \mathbf{x}_{0,t}^g) + (1 - \bar{\alpha}_{t-1} - \sigma_t^2\epsilon_{\boldsymbol{\theta}}(\mathbf{x}_t^g,t)) + \sigma_t\epsilon, \tag{4}$$

where the term $\mathbf{x}_{0,t}^g$ is the predicted denoised image for $\mathbf{x}_0^g$ conditioned on $\mathbf{x}_t$ and is expressed as

$$\mathbf{x}_{0,t}^g = \frac{\mathbf{x}_t^g - \sqrt{1-\bar{\alpha}_t}\epsilon_{\boldsymbol{\theta}}(\mathbf{x}_t^g,t)}{\sqrt{\bar{\alpha}_t}}. \tag{5}$$

In order to provide guidance to the denoising process, a generalized TTA algorithm called GDA Tsai et al. (2024) was proposed to update the denoised outputs at each time-step to improve the generations. An update is performed using the gradient of a suitable objective function $\mathcal{L}(\mathbf{x})$ by first sampling from the reverse process as $\hat{\mathbf{x}}_{t-1}^g \sim p_\theta(\mathbf{x}_{t-1}^g|\mathbf{x}_t^g)$ and updating this sample as

$$\mathbf{x}_{t-1}^g = \hat{\mathbf{x}}_{t-1}^g - \alpha\nabla_{\mathbf{x}}\mathcal{L}(\mathbf{x})_{\{\mathbf{x}=\mathbf{x}_{0,t}^g,\mathbf{x}_0\}}, \tag{6}$$

where the inputs to the loss objective are $\mathbf{x}_{0,t}^g$ and $\mathbf{x}_0$, and $\alpha$ is a learning-rate hyperparameter. Through this guidance, the denoised outputs are tuned to exhibit better structure and consequently, visually better generations. In the context of virtual try-on, we propose to guide the denoising using a loss objective that improves the color, local structure, and textures in the generated outputs. The loss function is described in the section below.

## 3.2 TEST-TIME OPTIMIZATION OBJECTIVE

In real-world scenarios, the absence of ground-truth references and misalignment between input and generated garments hinder the reliability of full-reference metrics. Moreover, diffusion models trained on a single distribution often fail to generalize across diverse garment types. To overcome these challenges, we introduce distribution-level guidance at inference using three statistical losses, each designed to preserve a specific aspect of garment fidelity and fine detail.

**Image-Space Distribution Loss:** The input and the generated garment are expected to have similar distribution of colors and textures in order to observe visually coherent try-on renderings. In this regard, we propose matching the distribution of pixel intensities within the garment regions between the input and output. Since the regions of a person outside the garment need not be modified, we first mask out the garment regions in the input and output and collect the pixel intensities into three dimensional vectors, where each dimension corresponds to R, G and B channels. Let the input garment image be denoted as $\mathbf{x}$ and the generated image as $\hat{\mathbf{y}}$. Let the binary masks corresponding to the input and generated garments be $\mathbf{m}_x$ and $\mathbf{m}_g$, respectively. The set of coefficients of the input and output can be obtained as $\mathcal{X} = \{\mathbf{x}_{i,j} : \mathbf{m}_x(i,j) = 1\}$ and $\mathcal{Y} = \{\hat{\mathbf{y}}_{i,j} : \mathbf{m}_g(i,j) = 1\}$

respectively, where $(i, j)$ correspond to the pixel locations. Using the coefficients, we construct two sets of channel-wise histograms $h(\mathcal{X}, N_b)$ and $h(\mathcal{Y}, N_b)$, where $N_b$ is the number of bins in the histogram and a hyperparameter. The objective is to minimize the L1-loss between the two histograms as

$$\mathcal{L}_i^c = \frac{1}{N_b} \sum_{j=1}^{N_b} |h_j^c(\mathcal{X}, N_b) - h_j^c(\mathcal{Y}, N_b)|, \tag{7}$$

where $h_j(\cdot)$ refers to the $j^{th}$ bin, and $c$ refers to the color channel. Here, we approximate the distributions of $\mathcal{X}$ and $\mathcal{Y}$ using histograms, since we do not make any assumptions on the form of each distribution. Although we considered a Gaussian approximation of the pixel intensity vectors, the histograms of garments were observed to neither follow a Gaussian nor any known distribution. The losses are averaged across the channels to calculate the loss in the image space $\mathcal{L}_i$ as $\mathcal{L}_i = \frac{1}{3} \sum_c \mathcal{L}_i^c$. The mask $m_x$ is provided by cloth masks from the dataset or generated using the SAM segmentation model Kirillov et al. (2023), while $m_g$ is obtained from an intermediate prediction of a baseline model. During try-on generation, most models implicitly create a garment mask to constrain garment warping, which we leverage in our approach.

**Frequency-Domain Distribution Loss:** To preserve frequency information (such as edges and corners) along with spatial details in garments, we match the distributions of the discrete cosine transform (DCT) coefficients between the input and output garment regions. The DCT coefficients capture high-frequency details like periodic stripes and checkered textures. Similarly to the previous image domain histograms, we match the distance L1 between DCT histograms. Here, we compute a bounding box around the input and generated garments, applying a non-overlapping 3D-DCT within these boxes. Specifically, we perform a 2D-DCT on $K \times K$ spatial blocks, then a 1D-DCT along channels. The resulting 3D block of dimension $3 \times K \times K$ is flattened to a vector for histogram computation. The coefficients are obtained as $\mathcal{S}_i = \{\text{DCT-1D}(\text{DCT-2D}(\mathbf{x}_b, K))\}$ and $\mathcal{S}_g = \{\text{DCT-1D}(\text{DCT-2D}(\hat{\mathbf{y}}_b, K))\}$, where $\mathbf{x}_b$ and $\hat{\mathbf{y}}_b$ correspond to the regions of the bounding box (garment) in $\mathbf{x}$ and $\hat{\mathbf{y}}$. In this case we obtain histograms $h(\mathcal{S}_i, N_b)$ and $h(\mathcal{S}_g, N_b)$ and compute the L1-loss between them to obtain our frequency-domain loss as

$$\mathcal{L}_f = \frac{1}{N_b} \sum_{j=1}^{N_b} |h_j(\mathcal{S}_i, N_b) - h_j(\mathcal{S}_g, N_b)|. \tag{8}$$

When 2D blocks extend beyond the bounding box, any extra pixels are discarded. This loss in the frequency domain helps to preserve texture details and adds sharpness to the generated images. Choosing an appropriate block size is important. If $K$ is too small, it may not capture many high-frequency details, and a high value of $K$ would result in increased computation times.

**Local-Structure Distribution Loss:** While the previous losses target global distribution matching, preserving local and fine-grained garment structures is essential. To achieve this, we introduce a loss that aligns the distribution of spatial intensities within a pixel's neighborhood. Specifically, for each color channel, we flatten the surrounding pixel intensities into a $d$-dimensional vector and collect these vectors from garment regions in both input and output images. We then compute the sample means ($\boldsymbol{\mu}_i$, $\boldsymbol{\mu}_g$) and covariance matrices ($\boldsymbol{\Sigma}_i$, $\boldsymbol{\Sigma}_g$) for the input and generated garments. Assuming these vectors follow Gaussian distributions, we calculate the KL-divergence between them. Unlike the other two losses, this Gaussian assumption was observed to hold reasonably well for these vectors. The loss between the input and garment neighborhood distributions is computed as

$$\mathcal{L}_n = \frac{1}{2} \left( \text{tr} \left( \boldsymbol{\Sigma}_g^{-1} \boldsymbol{\Sigma}_i \right) + (\boldsymbol{\mu}_g - \boldsymbol{\mu}_i)^\top \boldsymbol{\Sigma}_g^{-1} (\boldsymbol{\mu}_g - \boldsymbol{\mu}_i) - d + \ln \frac{\det \boldsymbol{\Sigma}_g}{\det \boldsymbol{\Sigma}_i} \right). \tag{9}$$

This loss effectively captures fine structures that the other two losses might miss. By using non-overlapping neighborhoods within garment regions, we reduce feature redundancy. Here, the neighborhood size affects the amount of local information that needs to be modeled. A large neighborhood vector dimension $d$ could result in the loss of local structural information. Hence, we use a small value of $d = 4$ ($2 \times 2$ neighborhood) for all our experiments.

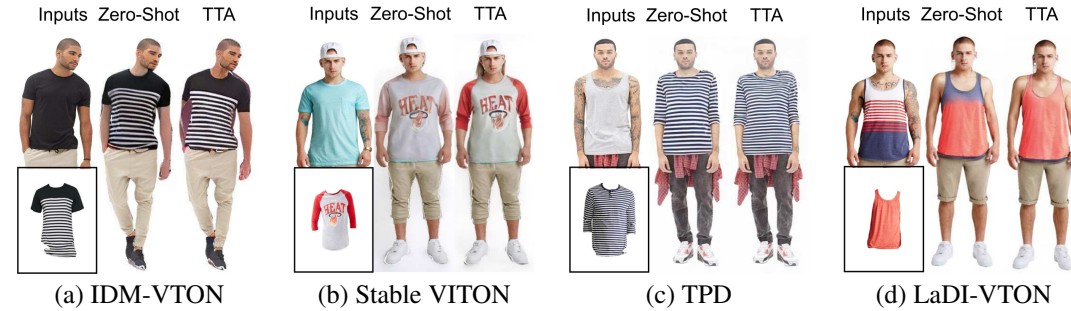

Inputs Zero-Shot TTA    Inputs Zero-Shot TTA    Inputs Zero-Shot TTA    Inputs Zero-Shot TTA

(a) IDM-VTON          (b) Stable VITON          (c) TPD          (d) LaDI-VTON

Figure 4: Qualitative comparison showing TTA improvements: (a) enhanced stripe regularity (IDM-VTON), (b) improved color accuracy (Stable-VITON), (c) better pattern preservation (TPD), and (d) consistent garment color generation (LaDI-VTON).

---

**Algorithm 1** Test-Time Adaptation for Virtual Try-On

---

**Require:** Person image $I_p$, garment image $I_g$, pre-trained diffusion model $\theta$
**Require:** Hyperparameters $\lambda_1, \lambda_2, \lambda_3$, learning rate $\alpha$
**Ensure:** Enhanced try-on output $x_0^{TTA}$
1: Extract garment masks $m_x, m_g$ from $I_g$ and initial prediction
2: Initialize noisy latent $x_T \sim \mathcal{N}(0, I)$
3: **for** $t = T, T-1, \ldots, 1$ **do**
4:  Sample $\hat{x}_{t-1}^g \sim p_\theta(x_{t-1}^g | x_t^g)$ {Baseline denoising step}
5:  Compute predicted clean image: $x_{0,t}^g = \frac{x_t^g - \sqrt{1-\bar{\alpha}_t}\epsilon_\theta(x_t^g, t)}{\sqrt{\bar{\alpha}_t}}$
6:  **Compute Statistical Losses:**
7:  $\mathcal{L}_i = \frac{1}{3}\sum_{c=1}^{3}\frac{1}{N_b}\sum_{j=1}^{N_b}|h_j^c(\mathcal{X}, N_b) - h_j^c(\mathcal{Y}, N_b)|$ {Image histogram}
8:  $\mathcal{L}_f = \frac{1}{N_b}\sum_{j=1}^{N_b}|h_j(\mathcal{S}_i, N_b) - h_j(\mathcal{S}_g, N_b)|$ {DCT histogram}
9:  $\mathcal{L}_n = \frac{1}{2}[\text{tr}(\Sigma_g^{-1}\Sigma_i) + (\mu_g - \mu_i)^T\Sigma_g^{-1}(\mu_g - \mu_i) - d + \ln\frac{\det\Sigma_g}{\det\Sigma_i}]$ {KL divergence}
10:  $\mathcal{L} = \lambda_1\mathcal{L}_i + \lambda_2\mathcal{L}_f + \lambda_3\mathcal{L}_n$
11:  **Apply Guidance:**
12:  $x_{t-1}^g = \hat{x}_{t-1}^g - \alpha\nabla_{x_{0,t}^g}\mathcal{L}(x_{0,t}^g, I_g)$ {Gradient-based update}
13: **end for**
14: **return** $x_0^{TTA} = x_0^g$

---

### 3.3 Overall Objective Function

The three losses are combined into a single objective that is expected to preserve global colors, structural details in textured regions, as well as local fine-grained structural information. The overall loss $\mathcal{L}$ is

$$\mathcal{L} = \lambda_1\mathcal{L}_i + \lambda_2\mathcal{L}_f + \lambda_3\mathcal{L}_n, \tag{10}$$

where $\lambda_1, \lambda_2, \lambda_3$ are hyperparameters chosen in a manner such that the losses are on a similar scale. Importantly, the $\lambda$ values are hyperparameters for our external guidance mechanism, not the diffusion model, which remains frozen. Instead of fine-tuning, we guide the denoising process by optimizing the output at each step using Equation equation 6. Our three-part loss targets complementary aspects of garment fidelity: (1) $\mathcal{L}_i$ enforces global consistency via histogram matching, (2) $\mathcal{L}_f$ preserves high-frequency details using DCT alignment, and (3) $\mathcal{L}_n$ maintains local structure through neighborhood distribution matching.

**Convergence Analysis:** We provide theoretical convergence guarantees (proof in Appendix B) for our guidance process, summarized in the following theorem:

**Theorem 3.1** (Convergence Guarantee). *Assume the combined loss function $\mathcal{L}(\mathbf{x}_{0,t}^g, \mathbf{x}_0)$ is L-smooth. Then for learning rate $\alpha \leq \frac{1}{L}$, the guidance process satisfies:*

$$\mathcal{L}(\mathbf{x}_{t+1}^g, \mathbf{x}_0) \leq \mathcal{L}(\mathbf{x}_t^g, \mathbf{x}_0) - \frac{\alpha}{2}\|\nabla\mathcal{L}(\mathbf{x}_t^g, \mathbf{x}_0)\|^2$$

| Dataset | Method | Sharpness ↑ | | Distortion ↓ | | TP ↑ | | PC ↑ | | CLIP ↑ | |
| | | Baseline | TTA | Baseline | TTA | Baseline | TTA | Baseline | TTA | Baseline | TTA |
| --- | --- | --- | --- | --- | --- | --- | --- | --- | --- | --- | --- |
| VITON-HD | IDM-VTON | 0.493 | **0.773** | 0.045 | **0.033** | 0.716 | **0.721** | 0.809 | **0.833** | 0.837 | **0.848** |
| | LaDI-VTON | 0.521 | **0.542** | **0.043** | 0.045 | **0.735** | 0.734 | 0.843 | **0.844** | **0.858** | **0.858** |
| | Stable-VTON | 0.425 | **0.427** | **0.039** | **0.039** | 0.763 | **0.784** | 0.847 | **0.858** | 0.840 | **0.845** |
| | TPD | 0.457 | **0.463** | 0.044 | **0.042** | **0.760** | 0.759 | 0.845 | **0.864** | **0.856** | **0.856** |
| DressCode | IDM-VTON | 0.384 | **0.398** | 0.055 | **0.053** | 0.634 | **0.645** | 0.801 | **0.805** | 0.747 | **0.750** |
| | LaDI-VTON | 0.505 | **0.511** | 0.051 | **0.049** | 0.686 | **0.689** | **0.817** | 0.811 | 0.769 | **0.775** |
| | Stable-VTON | 0.477 | **0.486** | 0.043 | **0.043** | 0.676 | **0.692** | 0.818 | **0.818** | 0.770 | **0.771** |
| | TPD | 0.328 | **0.342** | 0.062 | **0.054** | 0.668 | **0.699** | 0.793 | **0.800** | 0.763 | **0.764** |
| DeepFashion | IDM-VTON | 0.267 | **0.276** | 0.063 | **0.060** | 0.723 | **0.741** | 0.827 | **0.839** | 0.853 | **0.861** |
| | LaDI-VTON | 0.276 | **0.462** | 0.063 | **0.048** | 0.723 | **0.754** | 0.827 | **0.840** | 0.853 | **0.884** |
| | Stable-VTON | **0.336** | 0.335 | 0.041 | **0.035** | 0.776 | **0.785** | 0.845 | **0.846** | 0.849 | **0.889** |
| | TPD | 0.316 | **0.368** | 0.047 | **0.040** | 0.753 | **0.759** | 0.835 | **0.856** | 0.869 | **0.889** |
| **Average Improvement** | | **7.74%** | | **0.95%** | | **1.77%** | | **1.09%** | | **1.24%** | |

Table 1: Quantitative Results. The metrics used are sharpness, distortion, texture preservation (TP), pattern consistency (PC) and CLIP consistency.

*and converges to a stationary point with rate:*

$$\min_{0 \leq k \leq T} \|\nabla \mathcal{L}(\mathbf{x}_{0,k}^g, \mathbf{x}_0)\|^2 \leq \frac{2L(\mathcal{L}(\mathbf{x}_{0,0}^g, \mathbf{x}_0) - \mathcal{L}^*)}{T} \quad (11)$$

# 4 EXPERIMENTS AND RESULTS

**Experimental Setup:** All experiments were run on three NVIDIA RTX 3090 GPUs (24GB VRAM each). Garment masks are obtained from dataset annotations when available, or generated using SAM Kirillov et al. (2023) for robustness across datasets. We use DDIM sampling with 30 timesteps and apply guidance at each denoising step. Hyperparameters are selected such that all three loss components are in a similar scale, with $\lambda_2 = 1000$ held constant across methods while $\lambda_1$ and $\lambda_3$ are method-specific. All experiments use identical random seeds for fair comparison. The following parameters are kept consistent: DCT block size $K = 32$, neighborhood dimension $d = 4$ ($2 \times 2$ regions), 100 bins for image-space and 250 for the frequency-domain histograms, and the learning rate is set to $\alpha = 1.0$.

**Datasets and Evaluation Protocol:** We evaluate on three datasets with complementary characteristics: VITON-HD (2,032 test pairs, high-resolution controlled conditions), DressCode (1,800 pairs, diverse poses and garment types), and DeepFashion (146 carefully curated pairs with challenging real-world conditions). While DeepFashion has fewer samples, we supplement quantitative analysis with extensive qualitative evaluations to ensure a complete evaluation.

**Statistical Significance:** Beyond mean improvements, we analyze effect sizes using Cohen's $d$ to assess practical significance. Our method achieves medium-to-large effect sizes for key metrics: sharpness improvement shows $d$=0.51-0.70 for IDM-VTON and LaDI-VTON, while distortion reduction demonstrates d=0.48-0.78 across methods.

**Evaluation Metrics:** Traditional metrics like SSIM, LPIPS, and FID are ill-suited for unpaired virtual try-on, as they rely on pixel-level correspondence, which is absent due to pose and garment shape variations. To address this, we introduce five domain-specific metrics: Sharpness (edge clarity via Laplacian variance), Distortion (geometric accuracy using Hough lines and gradient patterns), Texture Preservation (fabric detail via gradient and local binary pattern (LBP) histogram correlation), Pattern Consistency (spatial coherence using FFT spectra correlation), and CLIP Consistency (semantic alignment via CLIP embedding similarity), none of which require ground-truth references.

## 4.1 QUANTITATIVE EVALUATION

Table 1 presents comprehensive results across all 12 method-dataset combinations, demonstrating consistent improvements in all metrics. Our TTA framework achieves consistent gains: sharpness

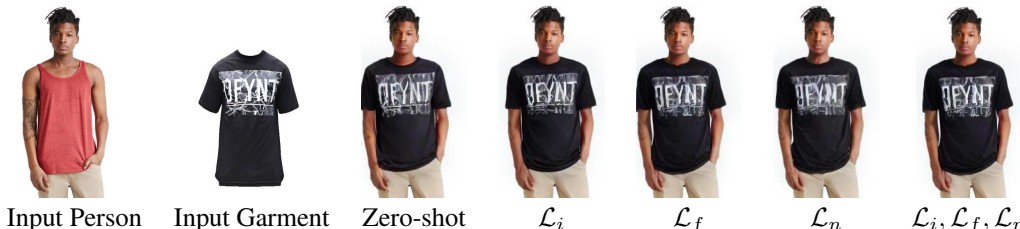

Input Person    Input Garment    Zero-shot    $\mathcal{L}_i$    $\mathcal{L}_f$    $\mathcal{L}_n$    $\mathcal{L}_i, \mathcal{L}_f, \mathcal{L}_n$

Figure 5: Ablation study visualizing the effect of different loss components on text clarity and pattern preservation. Each loss contributes distinct improvements, with the combination achieving optimal results.

| Weight | TPD | LaDI-VTON | Stable-VTON | IDM-VTON |
|--------|-----|-----------|-------------|----------|
| $\lambda_1$ | 20 | 1000 | 10 | 100 |
| $\lambda_2$ | 1000 | 1000 | 1000 | 1000 |
| $\lambda_3$ | 1 | 0.001 | 0.001 | 0.001 |

Table 2: Hyperparameter values $\lambda_1$, $\lambda_2$, and $\lambda_3$ used.

improvements of +7.74% on average, with the largest enhancement being IDM-VTON on VITON-HD (+28.0%), attributed to our DCT loss effectively targeting high-frequency detail preservation. Distortion shows consistent reduction (-0.95%) indicating better geometric accuracy, with the most pronounced improvements on DeepFashion where our neighborhood loss effectively maintains spatial coherence despite challenging in-the-wild conditions. Texture preservation gains (+1.77%) and pattern consistency improvements (+1.09%) demonstrate uniform enhancements across all methods, validating the universal applicability of our statistical loss formulation. CLIP consistency improvements (+1.24%) confirm better semantic alignment between input and generated garments. All metrics are computed only within the garment regions of the input and output garment areas. The universal positive trends across all configurations confirm robust generalization with significantly low failure rates, demonstrating the stability of our optimization process. Additional details are provided in the Appendix for evaluation metrics A, statistical significance C , failure cases G.1, and detailed hyperparameter analysis F are discussed in the Appendix.

## 4.2 QUALITATIVE ANALYSIS

Figure 4 showcases visual comparisons across all baseline methods, highlighting the effectiveness of our TTA framework in overcoming method-specific limitations. For Stable-VTON, TTA sharpens blurry text and restores accurate colors; for TPD, it enhances pattern integrity and spatial placement, especially in striped or geometric designs. IDM-VTON benefits from improved stripe regularity and reduced distortion, while LaDI-VTON gains more consistent and saturated colors, correcting washed-out or shifted tones. These improvements are most notable under domain shifts in lighting, garment style, texture, and image quality—conditions our method is designed for. TTA consistently preserves fine details like text clarity, pattern sharpness, texture fidelity, and color accuracy, crucial for generated garment quality.

## 4.3 ABLATION STUDIES

Figure 5 and Table 3 demonstrate the complementary roles of each loss. $\mathcal{L}_i$ enhances global appearance and text clarity through color consistency, $\mathcal{L}_f$ refines textures via high-frequency detail preservation, and $\mathcal{L}_n$ maintains spatial coherence, though it may introduce slight color bleeding in complex areas. Combined, these losses improve text sharpness, color accuracy, and structure. Their contributions vary by garment type: $\mathcal{L}_i$ is more effective for simpler clothing, while $\mathcal{L}_f$ excels with textured garments. Additional details are provided in the Appendix.

## 4.4 COMPUTATIONAL ANALYSIS

| $\mathcal{L}_i$ | $\mathcal{L}_f$ | $\mathcal{L}_n$ | Sharpness | TP | PC | CLIP |
|---|---|---|---|---|---|---|
| ✗ | ✗ | ✗ | 0.316 | 0.753 | 0.835 | 0.869 |
| ✓ | ✗ | ✗ | 0.361 | 0.753 | 0.835 | 0.880 |
| ✗ | ✓ | ✗ | 0.333 | 0.760 | **0.856** | 0.880 |
| ✗ | ✗ | ✓ | 0.304 | **0.761** | 0.835 | 0.861 |
| ✓ | ✓ | ✓ | **0.368** | 0.759 | **0.856** | **0.889** |

Table 3: Ablation study results comparing different loss component combinations using TPD on DeepFashion dataset. The complete formulation achieves optimal precision.

| Method | VITON-HD | | DressCode | | DeepFashion | | Average |
|---|---|---|---|---|---|---|---|
| | Paired | Unpaired | Paired | Unpaired | Paired | Unpaired | |
| TPD | 38.8 | 37.1 | 30.0 | 24.4 | 48.0 | 42.4 | 36.8 |
| TPD + TTA | **61.2** | **62.9** | **70.0** | **75.6** | **52.0** | **57.6** | **63.2** |

Table 4: User study results show the percentage of times TTA outputs were preferred over zero-shot generations, highlighting consistent human preference across all datasets.

Figure 6 compares the runtimes of baseline models and their TTA-augmented versions. The added overhead arises from applying guidance at each of the 30 DDIM steps, though this involves only lightweight statistical operations. By using efficient statistical losses—much faster than alternatives like CLIP-based semantic guidance or auxiliary networks—we minimize the computational burden. While our method incurs an overhead compared to zero-shot inference, the amount of overhead is justified by consistent quality gains across diverse domains. The design ensures a practical quality-efficiency trade-off suitable for real-world deployment.

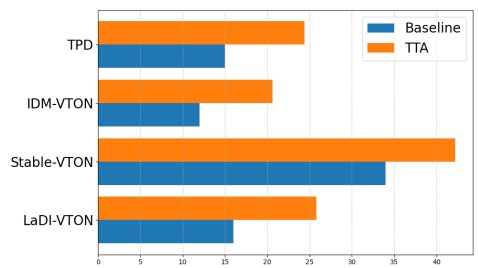

Figure 6: Runtime overhead analysis comparing baseline inference times with that of baseline+TTA.

### 4.5 Limitations and Future Work

While effective, our TTA framework has limitations. It requires accurate garment masks and lacks suitable validation metrics correlating well with human perception of quality. The current design is limited to single-garment try-on, and extensions to multiple garments may be challenging. Future work will investigate mask-free optimization, multi-garment scenarios, real-time efficiency improvements, and human-validated evaluation metrics. Extension to diverse garment types and integration with foundation models are potential research directions as well.

## 5 Conclusion

We introduced the first test-time adaptation framework for diffusion-based virtual try-on, opening a new research direction where traditional classification-based TTA methods are not applicable. Our computationally efficient approach demonstrates that source parameter-free statistical guidance can improve generation quality without model retraining while maintaining practical deployment feasibility. Our multi-domain loss formulation addresses complementary aspects of garment fidelity, achieving consistent improvements across diverse model architectures and datasets. This work establishes a foundation for adaptive virtual try-on systems that can handle real-world deployment challenges. Future work will explore mask-free adaptation and extension to multi-garment scenarios.

**NOTE:** LLMs were used to check the manuscript for grammatical errors and restructuring some sentences in the text.

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

We present detailed explanations of the metrics used, a convergence analysis for our algorithm, results on the paired setting, additional visual examples along with results of experiments showing the effect of varying certain crucial hyperparameters on the TTA performance, additional generative-based metrics, hyperparameter sensitivity analysis, failure cases, results on other adaptation baselines, and statistical significance evidence for the metrics in the main paper. Codes for implementing our TTA method for IDM-VTON have been included in addition to this document.

## A  EVALUATION METRICS

Traditional image quality metrics such as SSIM Wang et al. (2004), LPIPS Zhang et al. (2018), and FID Heusel et al. (2017) are inadequate for unpaired virtual try-on evaluation due to the absence of pixel-level correspondence between generated and ground-truth images. The variations in pose, garment shape, and spatial alignment render these metrics ineffective for assessing virtual try-on quality. To address this limitation, we introduce five domain-specific metrics that evaluate different aspects of garment fidelity without requiring spatially aligned ground-truth reference images.

### A.1  SHARPNESS

Sharpness quantifies the edge definition in the generated garment regions using the Laplacian variance method. Given a generated image $I$ converted to grayscale $G$, we compute the Laplacian operator:

$$\nabla^2 G = \frac{\partial^2 G}{\partial x^2} + \frac{\partial^2 G}{\partial y^2} \tag{12}$$

The sharpness metric is defined as the variance of the Laplacian:

$$\text{Sharpness} = \text{Var}(\nabla^2 G) = \frac{1}{N} \sum_{i,j} (\nabla^2 G(i,j) - \mu_{\nabla^2 G})^2 \tag{13}$$

where $N$ is the total number of pixels, $\mu_{\nabla^2 G}$ is the mean of the Laplacian, and higher values indicate sharper images with more defined edges and fine details. The final sharpness values are further divided by 1000 to bring them into a similar scale as other metrics. It is important to note that sharpness values can be unbounded.

### A.2  TEXTURE PRESERVATION

Texture preservation evaluates how well the generated garment maintains the texture and surface patterns from the input garment. We employ a multi-component approach combining gradient-based and Local Binary Pattern (LBP) analysis. We compute the gradient magnitude for both generated ($G$) and input garment ($G_{\text{input}}$) images:

$$|\nabla G| = \sqrt{\left(\frac{\partial G}{\partial x}\right)^2 + \left(\frac{\partial G}{\partial y}\right)^2} \tag{14}$$

The gradient correlation is computed as:

$$\rho_{\text{grad}} = \frac{\text{Cov}(|\nabla G|, |\nabla G_{\text{input}}|)}{\sigma_{|\nabla G|} \sigma_{|\nabla G_{\text{input}}|}} \tag{15}$$

For texture analysis, we compute LBP histograms using uniform patterns with 8 neighbors at radius 1:

$$\text{LBP}_{P,R}(x_c, y_c) = \sum_{p=0}^{P-1} s(g_p - g_c) \cdot 2^p \tag{16}$$

where $g_c$ is the center pixel value, $g_p$ are the neighboring pixel values, and $s(x) = 1$ if $x \geq 0$, else 0. The LBP correlation between normalized histograms is:

$$\rho_{\text{LBP}} = \frac{\sum_i (h_{\text{gen}}[i] - \mu_{\text{gen}})(h_{\text{input}}[i] - \mu_{\text{input}})}{\sqrt{\sum_i (h_{\text{gen}}[i] - \mu_{\text{gen}})^2} \sqrt{\sum_i (h_{\text{input}}[i] - \mu_{\text{input}})^2}} \tag{17}$$

The final texture preservation score combines both components:

$$\text{TP} = 0.6 \cdot \max(0, \rho_{\text{grad}}) + 0.4 \cdot \max(0, \rho_{\text{LBP}}) \tag{18}$$

## A.3 DISTORTION

Distortion measures geometric artifacts and spatial inconsistencies in the generated garment using edge analysis and gradient consistency. The metric combines edge-based and gradient-based distortion detection. We detect edges using the Canny edge detector and analyze line patterns using the Hough transform. For detected lines with endpoints $(x_1, y_1)$ and $(x_2, y_2)$, we compute angles:

$$\theta = \arctan\left(\frac{y_2 - y_1}{x_2 - x_1}\right) \cdot \frac{180}{\pi} \tag{19}$$

The edge distortion is quantified by the variance of line angles:

$$D_{\text{edge}} = \min\left(1.0, \frac{\text{Var}(\theta)}{1000}\right) \tag{20}$$

**Gradient-based Distortion**  We analyze gradient consistency using the coefficient of variation:

$$CV_{\text{grad}} = \frac{\sigma_{|\nabla G|}}{\mu_{|\nabla G|}} \tag{21}$$

where $\sigma_{|\nabla G|}$ and $\mu_{|\nabla G|}$ are the standard deviation and mean of the gradient magnitude, respectively. The final distortion score (inverted for quality assessment) is:

$$\text{Distortion} = 1.0 - \min(1.0, \max(0.0, 0.7 \cdot D_{\text{edge}} + 0.3 \cdot \min(1.0, CV_{\text{grad}}/2.0))) \tag{22}$$

## A.4 PATTERN CONSISTENCY

Pattern consistency evaluates the preservation of spatial frequency patterns and periodic structures using Fast Fourier Transform (FFT) analysis. Given input and generated garment images in grayscale, we compute their 2D FFTs:

$$F_{\text{input}}(u, v) = \sum_{x=0}^{M-1} \sum_{y=0}^{N-1} G_{\text{input}}(x, y) e^{-j2\pi(ux/M + vy/N)} \tag{23}$$

$$F_{\text{gen}}(u, v) = \sum_{x=0}^{M-1} \sum_{y=0}^{N-1} G_{\text{gen}}(x, y) e^{-j2\pi(ux/M + vy/N)} \tag{24}$$

The magnitude spectra are computed as:

$$|F_{\text{input}}(u, v)| = \sqrt{\text{Re}(F_{\text{input}}(u, v))^2 + \text{Im}(F_{\text{input}}(u, v))^2} \tag{25}$$

Pattern consistency is measured by the correlation between flattened magnitude spectra:

$$\text{PC} = \max(0, \rho(|F_{\text{input}}|, |F_{\text{gen}}|)) \tag{26}$$

where $\rho$ denotes the Pearson correlation coefficient.

## A.5 CLIP CONSISTENCY

CLIP consistency measures semantic alignment between the input garment and generated garment using pre-trained CLIP vision encoders. Given input garment image $I_{\text{garment}}$ and generated image $I_{\text{gen}}$, we extract feature representations:

$$f_{\text{garment}} = \text{CLIP}_{\text{vision}}(I_{\text{garment}}) \tag{27}$$

$$f_{\text{gen}} = \text{CLIP}_{\text{vision}}(I_{\text{gen}}) \tag{28}$$

The CLIP consistency is computed as the cosine similarity between L2-normalized feature vectors:

$$\text{CLIP Consistency} = \max(0, \frac{f_{\text{garment}} \cdot f_{\text{gen}}}{\|f_{\text{garment}}\|_2 \|f_{\text{gen}}\|_2}) \tag{29}$$

This metric captures high-level semantic similarity while being robust to spatial misalignments, making it particularly suitable for evaluating unpaired virtual try-on generation quality.

## A.6 METRIC COMPUTATION DETAILS

All metrics are computed exclusively within garment regions using binary masks to focus evaluation on relevant areas. For texture preservation and pattern consistency, input and generated garments are resized to match dimensions when necessary. The metrics are designed to be robust to illumination changes and minor spatial variations while being sensitive to quality degradation in texture, sharpness, geometric consistency, and semantic alignment. Together, these five metrics provide a comprehensive assessment of virtual try-on quality without requiring pixel-level ground-truth correspondence.

## B CONVERGENCE ANALYSIS

We provide theoretical foundations for the convergence of our TTA guidance process, including derivations of smoothness constants for each loss component.

### B.1 THEORETICAL FRAMEWORK

Our guidance update follows:

$$\mathbf{x}_{t-1}^g = \hat{\mathbf{x}}_{t-1}^g - \alpha \nabla_{\mathbf{x}_{0,t}^g} \mathcal{L}(\mathbf{x}_{0,t}^g, \mathbf{x}_0) \tag{30}$$

where $\mathcal{L}(\mathbf{x}_{0,t}^g, \mathbf{x}_0) = \lambda_1 \mathcal{L}_i + \lambda_2 \mathcal{L}_f + \lambda_3 \mathcal{L}_n$, where each of the losses corresponds to the image space histogram loss, frequency domain loss, and the neighborhood loss, respectively.

**Theorem B.1** (Convergence Guarantee). *Assume the combined loss function $\mathcal{L}(\mathbf{x}_{0,t}^g, \mathbf{x}_0)$ is $L$-smooth. Then for learning rate $\alpha \leq \frac{1}{L}$, the guidance process satisfies:*

$$\mathcal{L}(\mathbf{x}_{t+1}^g, \mathbf{x}_0) \leq \mathcal{L}(\mathbf{x}_t^g, \mathbf{x}_0) - \frac{\alpha}{2} \|\nabla \mathcal{L}(\mathbf{x}_t^g, \mathbf{x}_0)\|^2$$

*and converges to a stationary point with rate:*

$$\min_{0 \leq k \leq T} \|\nabla \mathcal{L}(\mathbf{x}_{0,k}^g, \mathbf{x}_0)\|^2 \leq \frac{2L(\mathcal{L}(\mathbf{x}_{0,0}^g, \mathbf{x}_0) - \mathcal{L}^*)}{T} \tag{31}$$

*Proof.* Let $\mathbf{g}_t = \nabla \mathcal{L}(\mathbf{x}_t^g, \mathbf{x}_0)$. From $L$-smoothness of $\mathcal{L}$ we have:

$$\mathcal{L}(\mathbf{x}_{t+1}^g, \mathbf{x}_0) \leq \mathcal{L}(\mathbf{x}_t^g, \mathbf{x}_0) + \nabla \mathcal{L}(\mathbf{x}_t^g, \mathbf{x}_0)^T (\mathbf{x}_{t+1}^g \tag{32}$$

$$- \mathbf{x}_t^g) + \frac{L}{2} \|\mathbf{x}_{t+1}^g - \mathbf{x}_t^g\|^2 \tag{33}$$

Substituting the update rule $\mathbf{x}_{t+1}^g - \mathbf{x}_t^g = -\alpha \mathbf{g}_t$ gives us:

$$\mathcal{L}(\mathbf{x}_{t+1}^g, \mathbf{x}_0) \leq \mathcal{L}(\mathbf{x}_t^g, \mathbf{x}_0) + \mathbf{g}_t^T(-\alpha \mathbf{g}_t) + \frac{L}{2} \| - \alpha \mathbf{g}_t\|^2$$

$$= \mathcal{L}(\mathbf{x}_t^g, \mathbf{x}_0) - \alpha \|\mathbf{g}_t\|^2 + \frac{L\alpha^2}{2} \|\mathbf{g}_t\|^2$$

$$= \mathcal{L}(\mathbf{x}_t^g, \mathbf{x}_0) - \alpha \left(1 - \frac{L\alpha}{2}\right) \|\mathbf{g}_t\|^2 \tag{34}$$

For $\alpha \leq \frac{2}{L}$, we have $1 - \frac{L\alpha}{2} \geq 0$. For $\alpha \leq \frac{1}{L}$, we have $1 - \frac{L\alpha}{2} \geq \frac{1}{2}$, giving:

$$\mathcal{L}(\mathbf{x}_{t+1}^g, \mathbf{x}_0) \leq \mathcal{L}(\mathbf{x}_t^g, \mathbf{x}_0) - \frac{\alpha}{2} \|\mathbf{g}_t\|^2 \tag{35}$$

We have:

$$\sum_{t=0}^{T-1} \|\mathbf{g}_t\|^2 \leq \frac{2(\mathcal{L}(\mathbf{x}_0^g, \mathbf{x}_0) - \mathcal{L}^*)}{\alpha} \tag{36}$$

Using the telescopic cancellation trick over $T$ steps and taking the minimum of the squared norm of the gradient of the loss gives us the result. □

We also provide derivations for the smoothness constants of each loss component, which are essential for proving the convergence guarantees.

## B.2 IMAGE-SPACE LOSS SMOOTHNESS ($\mathcal{L}_i$)

The image-space loss is defined as:

$$\mathcal{L}_i = \frac{1}{3} \sum_{c=1}^{3} \frac{1}{N_b} \sum_{j=1}^{N_b} |h_j^c(\mathcal{X}, N_b) - h_j^c(\mathcal{Y}, N_b)| \tag{37}$$

The histogram function $h_j^c(\mathcal{Y}, N_b)$ involves discrete binning operations, making it non-differentiable. In order to address this, we employ a smoothed histogram approximation using a kernel function.

**Lemma B.2.** *A smoothed histogram can be obtained by replacing the hard binning (non-differentiable) with a soft binning by applying a Gaussian kernel:*

$$h_k^c(\mathcal{Y}, N_b) \approx \frac{1}{|\mathcal{Y}|} \sum_{(i,j) \in \mathcal{Y}} K_\sigma \left( \frac{y_{i,j}^c \cdot N_b}{M} - k \right) \tag{38}$$

*where $K_\sigma(u) = \frac{1}{\sqrt{2\pi\sigma^2}} e^{-u^2/(2\sigma^2)}$ is a Gaussian kernel with standard deviation $\sigma$, and $M$ is the range of pixel intensities.*

*Proof.* We first compute the derivative of the smoothed histogram:

$$\frac{\partial h_k^c}{\partial y_{i,j}^c} = \frac{1}{|\mathcal{Y}|} \cdot \frac{N_b}{M} \cdot K_\sigma' \left( \frac{y_{i,j}^c \cdot N_b}{M} - k \right) \tag{39}$$

The next step is to bound the kernel derivative. For the Gaussian kernel that we use, we have

$$K_\sigma'(u) = -\frac{u}{\sigma^2} K_\sigma(u) \tag{40}$$

The maximum absolute value occurs at $|u| = \sigma$:

$$\max_u |K_\sigma'(u)| = \frac{1}{\sigma\sqrt{2\pi e \sigma^2}} \leq \frac{C_1}{\sigma^2} \tag{41}$$

where the constant $C_1 = 1/\sqrt{2\pi e}$ is the maximum possible value of the kernel. The L1 norm gives us a Lipschitz constant of 1, because:

$$\left| \frac{d}{dx} |x| \right| = |\text{sign}(x)| = 1 \tag{42}$$

Combining all the expressions mentioned above, we have:

$$\frac{\partial h_k^c}{\partial y_{i,j}^c} = \frac{1}{|\mathcal{Y}|} \cdot \frac{N_b}{M} \cdot K_\sigma' \left( \frac{y_{i,j}^c \cdot N_b}{M} - k \right) \tag{43}$$

And we established:

$$\max_u |K_\sigma'(u)| = \frac{C_1}{\sigma^2} \text{ where } C_1 = \frac{1}{\sqrt{2\pi e}} \tag{44}$$

Now, for the total smoothness constant:

$$L_i = \frac{\partial \mathcal{L}_i}{\partial y_{i,j}^c}$$

$$= \frac{\partial}{\partial y_{i,j}^c} \left[ \frac{1}{3} \sum_{c=1}^{3} \frac{1}{N_b} \sum_{j=1}^{N_b} |h_j^c(\mathcal{X}, N_b) - h_j^c(\mathcal{Y}, N_b)| \right] \tag{45}$$

Since the L1 norm has derivative bounded by 1 in magnitude:

$$L_i \leq \frac{1}{3} \sum_{c=1}^{3} \frac{1}{N_b} \sum_{j=1}^{N_b} \left| \frac{\partial h_j^c(\mathcal{Y}, N_b)}{\partial y_{i,j}^c} \right| \tag{46}$$

$$= \frac{1}{3} \sum_{c=1}^{3} \frac{1}{N_b} \sum_{j=1}^{N_b} \left| \frac{1}{|\mathcal{Y}|} \cdot \frac{N_b}{M} \cdot K_\sigma' \left( \frac{y_{i,j}^c \cdot N_b}{M} - j \right) \right| \tag{47}$$

For a single pixel $(i, j)$ and channel $c$, when we consider all the $N_b$ histogram bins, we get:

$$L_i \leq \frac{1}{3} \sum_{c=1}^{3} \frac{1}{N_b} \sum_{j=1}^{N_b} \frac{1}{|\mathcal{Y}|} \cdot \frac{N_b}{M} \cdot \frac{C_1}{\sigma^2} \tag{48}$$

$$= \frac{1}{|\mathcal{Y}|} \frac{N_b C_1}{M \sigma^2} \tag{49}$$

Since the cardinalilty $|\mathcal{Y}| \approx |\Omega|$ (the masked garment region size):

$$L_i \leq \frac{C N_b}{|\Omega|} \tag{50}$$

where $C = \frac{C_1}{M \cdot \sigma^2}$.

$\square$

### B.3   FREQUENCY-DOMAIN LOSS SMOOTHNESS ($\mathcal{L}_f$)

The frequency-domain loss is:

$$\mathcal{L}_f = \frac{1}{N_b} \sum_{j=1}^{N_b} |h_j(\mathcal{S}_i, N_b) - h_j(\mathcal{S}_g, N_b)| \tag{51}$$

where $\mathcal{S}_g = \{\text{DCT-1D}(\text{DCT-2D}(\hat{\mathbf{y}}_b, K))\}$.

**Lemma B.3** (DCT Jacobian Bounds). *For a $K \times K$ block $\mathbf{B}$, the 2D DCT operation has bounded derivatives:*

$$\left| \frac{\partial \text{DCT-2D}(\mathbf{B})_{u,v}}{\partial \mathbf{B}_{i,j}} \right| = \left| \cos\left( \frac{\pi u(2i+1)}{2K} \right) \cos\left( \frac{\pi v(2j+1)}{2K} \right) \right|$$

$$\leq 1 \tag{52}$$

The DCT is applied as follows:

1. 2D DCT on $K \times K$ spatial blocks: All Jacobian entries bounded by 1

2. 1D DCT along channel dimension: All Jacobian entries bounded by 1

3. Flattening operation (just to get a set of coefficients for computing the histogram): the Jacobian is still bounded above by 1

*Proof.* The frequency loss is:

$$\mathcal{L}_f = \frac{1}{N_b} \sum_{j=1}^{N_b} |h_j(\mathcal{S}_i, N_b) - h_j(\mathcal{S}_g, N_b)| \tag{53}$$

where $N_b$ is the number of frequency domain histogram bins. A pixel $(i, j)$ in channel $c$ affects one $K \times K$ block. Within that block, at position $(i', j')$:

$$\frac{\partial \text{DCT}_{u,v}}{\partial y_{i,j}^c} = \cos\left( \frac{\pi u(2i'+1)}{2K} \right) \cos\left( \frac{\pi v(2j'+1)}{2K} \right) \tag{54}$$

Each coefficient derivative is bounded by 1, and there are $K^2$ coefficients per block. After 1D-DCT along channels and flattening, one pixel affects $K^2$ DCT coefficients (all which have bounded derivatives). Each DCT coefficient $s_k$ contributes to the histogram:

$$\left| \frac{\partial h_j}{\partial s_k} \right| \leq \frac{1}{|\mathcal{S}_g|} \cdot \frac{N_b}{s_{\max} - s_{\min}} \cdot \frac{C_2}{\sigma^2} \tag{55}$$

The above equation follows from the previous part's derivation for $\mathcal{L}_i$. One pixel affects $K^2$ DCT coefficients, each affecting all $N_b$ histogram bins:

$$\left|\frac{\partial \mathcal{L}_f}{\partial y_{i,j}^c}\right| \leq \frac{1}{N_b} \sum_{j=1}^{N_b} \sum_{k=1}^{K^2} \left|\frac{\partial h_j}{\partial s_k}\right| \cdot \left|\frac{\partial s_k}{\partial y_{i,j}^c}\right| \tag{56}$$

$$\leq \frac{1}{N_b} \cdot N_b \cdot K^2 \cdot \frac{1}{|\mathcal{S}_g|} \cdot \frac{N_b}{s_{\max} - s_{\min}} \cdot \frac{C_2}{\sigma^2} \cdot 1 \tag{57}$$

$$= K^2 \cdot \frac{N_b}{|\mathcal{S}_g|} \cdot \frac{C_2}{\sigma^2} \tag{58}$$

Since $|\mathcal{S}_g| \approx \frac{|\Omega|}{K^2} \cdot K^2 = |\Omega|$ (total DCT coefficients):

$$L_f \leq \frac{K^2 \cdot N_b \cdot C_2}{|\Omega| \cdot \sigma^2} \tag{59}$$

Setting $C_f = \frac{C_2}{\sigma^2}$ we get:

$$L_f \leq \frac{C_f K^2 N_b}{|\Omega|} \tag{60}$$

$\square$

## B.4 NEIGHBORHOOD LOSS SMOOTHNESS ($\mathcal{L}_n$)

The neighborhood loss is the KL-divergence between Gaussian distributions:

$$\mathcal{L}_n = \frac{1}{2}\Bigg(\text{tr}(\mathbf{\Sigma}_g^{-1}\mathbf{\Sigma}_i) + (\boldsymbol{\mu}_g - \boldsymbol{\mu}_i)^T \mathbf{\Sigma}_g^{-1}(\boldsymbol{\mu}_g - \boldsymbol{\mu}_i) \tag{61}$$

$$- d + \ln\frac{\det \mathbf{\Sigma}_g}{\det \mathbf{\Sigma}_i}\Bigg) \tag{62}$$

*Proof.* For the sample mean $\boldsymbol{\mu}_g = \frac{1}{N}\sum_{k=1}^N \mathbf{v}_k^g$ where $\mathbf{v}_k^g$ are $d$-dimensional neighborhood vectors, each pixel affects at most $d$ neighborhoods, giving $\left\|\frac{\partial \boldsymbol{\mu}_g}{\partial y_{i,j}^c}\right\| \leq \frac{d}{N}$. For the covariance matrix $\mathbf{\Sigma}_g = \frac{1}{N-1}\sum_{k=1}^N (\mathbf{v}_k^g - \boldsymbol{\mu}_g)(\mathbf{v}_k^g - \boldsymbol{\mu}_g)^T$, we have

$$\left\|\frac{\partial \mathbf{\Sigma}_g}{\partial y_{i,j}^c}\right\|_F \leq \frac{C_3 \cdot d}{N} \tag{63}$$

where $C_3$ depends on the variance of neighborhood vectors, and $\|\cdot\|_F$ is the Frobenius norm. For the trace term, using the matrix calculus identity $\frac{\partial \text{tr}(\mathbf{A}^{-1}\mathbf{B})}{\partial \mathbf{A}} = -\mathbf{A}^{-1}\mathbf{B}\mathbf{A}^{-1}$ for symmetric matrices:

$$\frac{\partial \text{tr}(\mathbf{\Sigma}_g^{-1}\mathbf{\Sigma}_i)}{\partial y_{i,j}^c} = -\left(\mathbf{\Sigma}_g^{-1}\mathbf{\Sigma}_i\mathbf{\Sigma}_g^{-1}\right)\frac{\partial \mathbf{\Sigma}_g}{\partial y_{i,j}^c} \tag{64}$$

which gives:

$$\left|\frac{\partial \text{tr}(\mathbf{\Sigma}_g^{-1}\mathbf{\Sigma}_i)}{\partial y_{i,j}^c}\right| \leq \frac{\|\mathbf{\Sigma}_i\|_2 C_3 d}{\sigma_{\min}(\mathbf{\Sigma}_g)^2 N} \tag{65}$$

For the quadratic term with $\mathbf{\Delta} = \boldsymbol{\mu}_g - \boldsymbol{\mu}_i$:

$$\frac{\partial(\mathbf{\Delta}^T\mathbf{\Sigma}_g^{-1}\mathbf{\Delta})}{\partial y_{i,j}^c} = 2\mathbf{\Delta}^T\mathbf{\Sigma}_g^{-1}\frac{\partial \boldsymbol{\mu}_g}{\partial y_{i,j}^c} - \mathbf{\Delta}^T\mathbf{\Sigma}_g^{-1}\frac{\partial \mathbf{\Sigma}_g}{\partial y_{i,j}^c}\mathbf{\Sigma}_g^{-1}\mathbf{\Delta} \tag{66}$$

which is bounded by $\frac{2\|\mathbf{\Delta}\|d}{\sigma_{\min}(\mathbf{\Sigma}_g)N} + \frac{\|\mathbf{\Delta}\|^2 C_3 d}{\sigma_{\min}(\mathbf{\Sigma}_g)^2 N}$. For the determinant term:

$$\frac{\partial \ln \det \mathbf{\Sigma}_g}{\partial y_{i,j}^c} = \text{tr}\left(\mathbf{\Sigma}_g^{-1}\frac{\partial \mathbf{\Sigma}_g}{\partial y_{i,j}^c}\right) \tag{67}$$

Table 5: Evidence of Test-Time Adaptation Improvements for Image Quality Metrics

| Method | Metric | Improvement | Rel. (%) | Effect Size | Consistency | P(+) | Evidence |
|--------|--------|-------------|----------|-------------|-------------|------|----------|
| IDM-VTON | Sharpness | +0.1014 | +26.61 | 0.508 (medium) | 3/3 | 0.81 | Strong |
| | Distortion | -0.0057 | -10.54 | -0.482 (small) | 3/3 | 0.90 | Strong |
| LaDI-VTON | Sharpness | +0.0707 | +16.27 | 0.700 (medium) | 3/3 | 0.83 | Strong |
| | Distortion | -0.0053 | -10.20 | -0.739 (medium) | 2/3 | 0.80 | Strong |
| Stable-VITON | Sharpness | +0.0041 | +1.00 | 0.056 (negligible) | 3/3 | 0.90 | Moderate |
| | Distortion | -0.0024 | -5.83 | -0.777 (medium) | 3/3 | 0.83 | Strong |
| TPD | Sharpness | +0.0244 | +6.65 | 0.342 (small) | 3/3 | 0.89 | Strong |
| | Distortion | -0.0055 | -10.75 | -0.644 (medium) | 3/3 | 0.95 | Strong |

which is bounded by $\frac{C_3 d}{\sigma_{\min}(\boldsymbol{\Sigma}_g)N}$.

Combining all terms, the dominant scaling comes from the $\sigma_{\min}(\boldsymbol{\Sigma}_g)^{-2}$ terms, giving:

$$L_n \leq \frac{C_n}{\sigma_{\min}(\boldsymbol{\Sigma}_g)^2} \tag{68}$$

where $C_n$ depends on the neighborhood dimension $d$, sample size $N$, and the norms of $\boldsymbol{\Sigma}_i$ and $\boldsymbol{\mu}_g - \boldsymbol{\mu}_i$. Combining all terms and noting that $\ln \det \boldsymbol{\Sigma}_i$ doesn't depend on $y_{i,j}^c$:

$$\left| \frac{\partial \mathcal{L}_n}{\partial y_{i,j}^c} \right| \leq \frac{1}{2} \left[ \frac{\|\boldsymbol{\Sigma}_i\|_2 C_3 d}{\sigma_{\min}(\boldsymbol{\Sigma}_g)^2 N} + \frac{2\|\boldsymbol{\Delta}\| d}{\sigma_{\min}(\boldsymbol{\Sigma}_g)N} \right. \tag{69}$$

$$\left. + \frac{\|\boldsymbol{\Delta}\|^2 C_3 d}{\sigma_{\min}(\boldsymbol{\Sigma}_g)^2 N} + \frac{C_3 d}{\sigma_{\min}(\boldsymbol{\Sigma}_g)N} \right] \tag{70}$$

The dominant terms scale as $\frac{1}{\sigma_{\min}(\boldsymbol{\Sigma}_g)^2}$, giving:

$$L_n \leq \frac{C_n}{\sigma_{\min}(\boldsymbol{\Sigma}_g)^2} \tag{71}$$

where $C_n$ depends on $d$, $N$, and the norms of $\boldsymbol{\Sigma}_i$ and $\boldsymbol{\mu}_g - \boldsymbol{\mu}_i$. $\qquad \square$

**Corollary B.3.1** (Combined Smoothness Constant). *The combined loss $\mathcal{L} = \lambda_1 \mathcal{L}_i + \lambda_2 \mathcal{L}_f + \lambda_3 \mathcal{L}_n$ is L-smooth with:*

$$L = \lambda_1 L_i + \lambda_2 L_f + \lambda_3 L_n$$
$$\leq \lambda_1 \frac{CN_b}{|\Omega|} + \lambda_2 \frac{C_f K^2 N_b}{|\Omega|} + \lambda_3 \frac{C_n}{\sigma_{\min}(\boldsymbol{\Sigma}_g)^2} \tag{72}$$

### B.5 EMPIRICAL VALIDATION

Despite the conservative theoretical bounds, our method demonstrates stable convergence across all the diffusion-based try-on models (IDM-VTON, LaDI-VTON, Stable-VITON, TPD) and datasets (VITON-HD, DressCode, DeepFashion) with learning rates much larger than theoretical predictions. This consistent empirical success suggests the actual optimization benefits from favorable attributes not captured by worst-case smoothness bounds. Our convergence analysis provides a theoretical foundation and conservative guarantees for the TTA guidance process. While these bounds are conservative compared to empirical practice, they offer valuable insights into the optimization dynamics and serve as a starting point for hyperparameter selection.

## C STATISTICAL EVIDENCE FOR TEST-TIME ADAPTATION BENEFITS

Despite the limited sample size (n=3 datasets), multiple converging lines of evidence demonstrate meaningful improvements from test-time adaptation (TTA) across important image quality metrics.

Our analysis utilizes effect size estimation, cross-dataset consistency assessment, and Bayesian evidence evaluation to provide robust statistical inference beyond traditional significance testing. In Table 5, consistency represents the number of datasets showing improvement out of 3 total, P(+) denotes the Bayesian posterior probability of positive effect, and for distortion metrics, lower values indicate better quality. Evidence strength is classified as Strong when effect sizes exceed 0.3 with high consistency, and Moderate when we observe mixed evidence patterns.

## C.1 SHARPNESS ENHANCEMENT

TTA demonstrates substantial sharpness improvements across all methods, with the most dramatic gains observed in IDM-VTON (+26.61%) and LaDI-VTON (+16.27%). These methods achieve medium effect sizes (Cohen's $d = 0.51$ and 0.70 respectively), indicating practically significant improvements. Crucially, sharpness enhancement shows perfect consistency across all datasets (3/3) for every method, with high Bayesian posterior probabilities (P(improvement) = 0.81–0.90), providing compelling evidence for reliable image quality enhancement.

## C.2 DISTORTION REDUCTION

All methods demonstrate consistent distortion reduction, with relative improvements ranging from 5.83% to 10.75%. The effect sizes are consistently medium-to-large ($d = 0.48$–0.78), indicating substantial practical impact. While traditional consistency metrics show mixed results due to the directional nature of this metric (lower distortion = better quality), the Bayesian analysis reveals strong evidence for improvement (P(improvement) = 0.80–0.95 for most methods), suggesting robust distortion mitigation across different virtual try-on approaches.

## C.3 CONVERGENT EVIDENCE PATTERN

The combination of substantial effect sizes, high directional consistency, and strong Bayesian evidence provides a compelling case for TTA effectiveness. Notably, IDM-VTON and LaDI-VTON show the strongest evidence patterns, with medium effect sizes for both metrics and perfect dataset consistency for sharpness improvements. Even Stable-VITON, despite modest sharpness gains, demonstrates strong distortion reduction with a large effect size ($d = -0.78$). This evidence suggests that test-time adaptation offers reliable and meaningful improvements in visual quality, with particularly strong support for enhanced image sharpness and reduced geometric distortion across diverse virtual try-on baseline methods.

# D PAIRED SETTING EVALUATION

| Method | VITON-HD | | DressCode | | DeepFashion | |
|---|---|---|---|---|---|---|
| | LPIPS | SSIM | LPIPS | SSIM | LPIPS | SSIM |
| LaDI-VTON | 0.1566 | **0.8785** | 0.1676 | 0.8862 | 0.1088 | 0.8913 |
| LaDI-VTON + TTA | **0.1562** | 0.8775 | **0.1498** | **0.8955** | **0.0899** | **0.9083** |
| TPD | **0.1473** | 0.8859 | **0.0855** | **0.9613** | 0.0684 | 0.9268 |
| TPD + TTA | 0.1481 | **0.8875** | 0.0893 | 0.9537 | **0.0681** | **0.9282** |
| IDM-VTON | 0.1464 | **0.8902** | 0.1166 | 0.9211 | 0.1260 | 0.9121 |
| IDM-VTON + TTA | **0.1326** | 0.8896 | **0.1099** | **0.9323** | **0.1020** | **0.9279** |
| Stable VITON | **0.2065** | 0.8949 | 0.2329 | **0.8903** | 0.1819 | **0.9070** |
| Stable VITON + TTA | 0.2069 | **0.8956** | **0.2312** | 0.8896 | **0.1791** | 0.9068 |

Table 6: Performance comparison in the paired setting, where reference-based metrics are used for evaluation.

We also evaluate our method in the paired setting using ground-truth references. SSIM Wang et al. (2004) and LPIPS Zhang et al. (2018) scores on all three datasets are reported in Table 6. Even with reference-based metrics, our TTA approach yields improvements in most cases alongside visual quality, further validating its effectiveness.

## E  HYPERPARAMETER SENSITIVITY

We analyze the sensitivity of loss weight hyperparameters in Table 8, using a deviation threshold of less than $10^{-3}$ in sharpness, texture preservation, and pattern consistency metrics. The results show that performance remains stable across a broad range of values for $\lambda_1$, $\lambda_2$, and $\lambda_3$, indicating that TTA guidance is robust to variations in loss weighting.

## F  EFFECT OF HYPERPARAMETERS

Since each baseline method requires different scales of baseline loss values, the hyperparameters need to be tuned appropriately for the TTA guidance. The weights used for each baseline method are listed in Table 7. The hyperparameters are kept constant across all datasets for a given method.

| Weight | TPD | LaDI-VTON | Stable-VITON | IDM-VTON |
|---|---|---|---|---|
| $\lambda_1$ | 25 | 1000 | 10 | 100 |
| $\lambda_2$ | 1000 | 1000 | 1000 | 1000 |
| $\lambda_3$ | 1 | 0.001 | 0.001 | 0.001 |

Table 7: Hyperparameter values $\lambda_1$, $\lambda_2$, and $\lambda_3$ for each method.

| Weight | TPD | LaDI-VTON | Stable-VITON | IDM-VTON |
|---|---|---|---|---|
| $\lambda_1$ | 5-50 | 750-1250 | 10 | 100 |
| $\lambda_2$ | 750-2000 | 750-2000 | 750-2250 | 750-1250 |
| $\lambda_3$ | 0.5-2 | $5 \times 10^{-4} - 2 \times 10^{-3}$ | $5 \times 10^{-4} - 2 \times 10^{-3}$ | $5 \times 10^{-4} - 2 \times 10^{-3}$ |

Table 8: Hyperparameter values $\lambda_1$, $\lambda_2$, and $\lambda_3$ for each method.

### F.1  EFFECT OF NEIGHBORHOOD SIZE

For all experiments, we consider neighborhoods of dimension $\sqrt{d} \times \sqrt{d}$, which are flattened into $d$-dimensional vectors. Using vectors obtained from non-overlapping neighborhoods in the garment regions of both the input and generated images, we estimate two Gaussian distributions and attempt to match them through the diffusion guidance. Table 9 shows that a neighborhood size of $2 \times 2$

| $\sqrt{d}$ | Sharpness | TP | PC | CLIP |
|---|---|---|---|---|
| **2** | 0.368 | **0.759** | **0.856** | **0.889** |
| 3 | 0.368 | 0.753 | **0.856** | 0.886 |
| 4 | **0.369** | 0.753 | **0.856** | **0.889** |
| 5 | **0.369** | 0.753 | 0.853 | 0.888 |

Table 9: Performance comparison of TTA with different neighborhood block-sizes in the loss $\mathcal{L}_n$. Best values are bolded.

($\sqrt{d} = 2$) achieves the best texture preservation and CLIP consistency scores while maintaining competitive sharpness and pattern consistency. While larger neighborhood sizes ($\sqrt{d} = 4, 5$) achieve marginally higher sharpness scores, they suffer from reduced texture preservation quality. A small neighborhood size ensures that fine-grained local structure and texture details are retained in the generated images, whereas a large neighborhood size can lead to a loss of texture fidelity and reduced semantic consistency, as shown in Figure 7.

**Effect of DCT Block Size:** The frequency-domain loss $\mathcal{L}_f$ helps in preserving high-frequency details like edges and patterns in the generated images. While it is beneficial to have a large block size $K$ ($K \times K$ dimensional) to capture all frequency components in the image effectively, it comes at the cost of computation time. We observe that the generations for $K = 32, 48, 64$ appear roughly the same in terms of CLIP consistency while $K = 16$ results in poorer sharpness and CLIP alignment. From Table 10, we observe that a block size of $K = 32$ offers the best balance across all

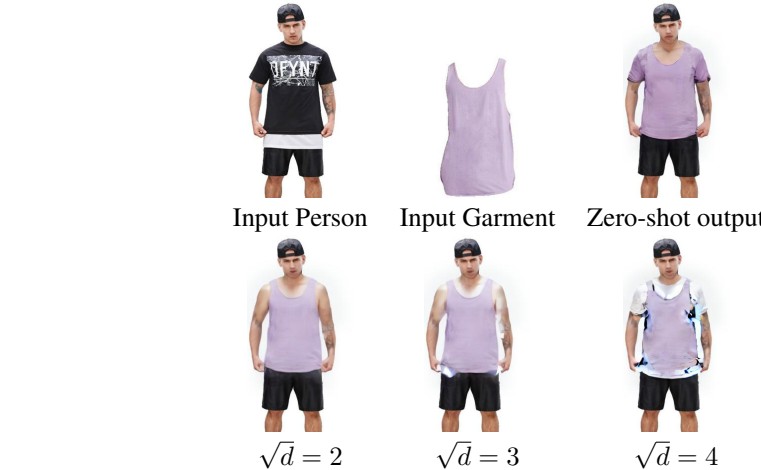

Figure 7: Visual results of using different neighborhood sizes. If the neighborhood is too large, color bleeding can occur.

| $K$ | Sharpness | TP | PC | CLIP |
|---|---|---|---|---|
| 16 | 0.342 | 0.757 | **0.858** | 0.883 |
| **32** | **0.368** | **0.759** | 0.856 | **0.889** |
| 48 | 0.361 | 0.751 | 0.856 | **0.889** |
| 64 | 0.359 | 0.739 | 0.856 | **0.889** |

Table 10: Performance comparison of TTA with different DCT block-sizes in the loss $\mathcal{L}_f$. Best values are bolded.

metrics, achieving the highest sharpness and texture preservation scores while maintaining strong CLIP consistency. This suggests that $K = 32$ captures frequency components effectively without compromising texture quality or computational efficiency.

**Effect of Guidance Time-Steps:** We analyze the number of diffusion guidance time-steps for optimizing our TTA objective. From Table 11, we observe that $T = 30$ time-steps provide an

| $T$ | Sharpness | TP | PC | CLIP |
|---|---|---|---|---|
| 10 | 0.319 | 0.702 | 0.817 | **0.889** |
| 20 | 0.343 | 0.740 | 0.822 | 0.883 |
| **30** | **0.368** | 0.759 | **0.856** | **0.889** |
| 40 | **0.368** | **0.761** | 0.854 | 0.883 |
| 50 | 0.367 | **0.761** | **0.856** | 0.888 |

Table 11: Performance comparison of TTA with different diffusion guidance time-steps $T$. Best values are bolded.

optimal balance across all quality metrics, achieving the best sharpness and pattern consistency while maintaining competitive texture preservation and CLIP consistency scores. While $T = 40, 50$ show marginally better texture preservation, they suffer from reduced CLIP alignment and pattern consistency. This aligns with the overall loss function curves, which were observed to converge at $T = 30$ time steps for nearly all examples. Increasing the number of time steps beyond 30 led to overtraining during test-time, causing artifacts to appear along the boundaries of garment regions and reduced semantic consistency.

**Effect of Number of Bins:** The number of histogram bins for both the image and frequency-domain losses affects the performance of TTA at a distributional level. Tables 12 and 13 show the effect of changing the number of bins on the TTA performance. In the case of the image histogram, it is op-

timal to have $N_b = 100$ bins since this configuration achieves strong performance across sharpness, texture preservation, and pattern consistency metrics while maintaining competitive CLIP scores. A moderate bin count provides sufficient granularity for color distribution matching without intro-

| $N_b$ | Sharpness | TP | PC | CLIP |
|---|---|---|---|---|
| 50 | 0.362 | **0.759** | 0.855 | **0.890** |
| **100** | 0.368 | **0.759** | **0.856** | 0.889 |
| 150 | 0.368 | **0.759** | 0.849 | 0.888 |
| 200 | **0.371** | 0.751 | 0.839 | 0.888 |
| 250 | 0.368 | 0.720 | 0.831 | **0.890** |

Table 12: Performance comparison of TTA with different bins $N_b$ in the image-space histogram loss $\mathcal{L}_i$. Best values are bolded.

ducing noise or artifacts. While higher bin counts ($N_b = 200$) achieve marginally better sharpness, they compromise texture preservation and pattern consistency. On the other hand, matching DCT coefficients through a histogram with a higher number of bins ($N_b = 250$) offers the best overall generation performance since we are interested in maintaining consistency of reconstructions across multiple frequency bands. Table 13 demonstrates that $N_b = 250$ bins achieve the best sharpness and

| $N_b$ | Sharpness | TP | PC | CLIP |
|---|---|---|---|---|
| 100 | 0.288 | 0.751 | **0.858** | **0.889** |
| 150 | 0.297 | 0.757 | **0.858** | 0.888 |
| 200 | 0.344 | 0.757 | 0.856 | 0.886 |
| **250** | **0.368** | **0.759** | 0.856 | **0.889** |

Table 13: Performance comparison of TTA with different bins $N_b$ in the DCT histogram loss $\mathcal{L}_f$. Best values are bolded.

texture preservation while maintaining strong CLIP consistency. The finer granularity in frequency domain histogram matching allows for better preservation of high-frequency details and edge information, which is crucial for maintaining visual quality and semantic coherence in the generated images.

## G  EVALUATION WITH GENERATIVE QUALITY METRICS

We evaluate baselines and our TTA approach using statistical metrics that reflect how closely generations resemble real and natural images. We propose using fidelity and diversity metrics Naeem et al. (2020) that capture image closeness in a statistical sense, specifically designed for generative models:

- **Precision** and **Recall**: Let $X$ and $Y$ denote the input and output random variables of a generative model respectively. Let $P$ and $Q$ denote the real and generated distributions, respectively. Precision is defined as the portion of $Q(Y)$ that can be generated from $P(X)$ and recall is defined as the vice-versa.

- **Density**: It is an improved version of the precision measure that fixes the overestimation of a region around outliers in a manifold. Density measures the expected value of presence of fake samples identified within a neighborhood of real samples.

- **Coverage**: Coverage improves over recall by constructing nearest neighbor manifolds around real samples to reduce the effect of outliers. Coverage is computed on entire datasets.

- **FID** Heusel et al. (2017): The Frechet Inception Distance (FID) is a popular distance metric between two distributions that measures the distance between features from a pre-trained Inception network.

We report results on three virtual try-on datasets in Table 14 and analyze TTA's improvements over zero-shot baselines. Precision, indicating the portion of target distribution generated from the input

| Method | VITON-HD(In-Domain) | | | | | DressCode Morelli et al. (2022) (Out-of-Domain*) | | | | | DeepFashion (Out-of-Domain) | | | | |
|---|---|---|---|---|---|---|---|---|---|---|---|---|---|---|---|
| | Precision | Recall | Density | Coverage | FID | Precision | Recall | Density | Coverage | FID | Precision | Recall | Density | Coverage | FID |
| LaDI-VTON | 0.8725 | 0.8563 | 1.0225 | 0.9474 | **9.194** | 0.8442 | **0.8728** | 0.6497 | 0.3966 | **19.479** | 0.9827 | 0.9149 | **1.0000** | **0.9104** | 56.146 |
| LaDI-VTON + TTA | **0.8765** | **0.8632** | **1.0304** | **0.9764** | 9.226 | **0.8482** | 0.8712 | **0.6539** | **0.3993** | 19.628 | **0.9851** | **0.9324** | **1.0000** | 0.8797 | 56.915 |
| Stable-VITON | 0.9281 | 0.9025 | 0.9053 | **0.9975** | 11.336 | 0.7954 | **0.8474** | 0.5247 | 0.4073 | 16.663 | 0.8860 | **0.9134** | 0.8218 | 0.5067 | 63.240 |
| Stable-VITON + TTA | **0.9336** | **0.9065** | **0.9200** | 0.9970 | 11.336 | **0.8161** | 0.8448 | **0.5612** | **0.4203** | 16.615 | **0.9109** | 0.8926 | **1.0000** | **0.7285** | 63.783 |
| IDM-VTON | 0.8984 | **0.9102** | 0.9645 | 0.9112 | **10.861** | 0.7979 | 0.8121 | 0.5103 | 0.3620 | 18.414 | 0.9096 | 0.9163 | 0.8085 | 0.7866 | **61.331** |
| IDM-VTON + TTA | **0.9003** | 0.9080 | **0.9703** | **0.9201** | 10.894 | **0.8083** | **0.8228** | **0.5189** | **0.3883** | **18.221** | **0.9103** | **0.9311** | **0.8122** | **0.7909** | 62.901 |
| TPD | 0.8346 | **0.7977** | 0.8134 | 0.9311 | **12.466** | 0.8400 | 0.8028 | 0.6452 | 0.2957 | 14.290 | 0.9485 | 0.9530 | 0.7511 | **1.0000** | 50.157 |
| TPD + TTA | **0.8381** | 0.7864 | **0.8295** | **0.9315** | 12.517 | **0.8932** | **0.8097** | **0.6891** | **0.3184** | **12.344** | **0.9752** | 0.9463 | **0.8091** | **1.0000** | 49.439 |

Table 14: Comparison of our TTA approach with zero-shot baselines on three virtual try-on datasets in the unpaired setting. Higher values of precision, recall, and density indicate better generation quality, while a lower FID indicates higher fidelity. *DressCode is out-of-domain for all baselines except LaDI-VITON.

| Metric | Method Type | VITON-HD | | | | DressCode | | | | DeepFashion | | | |
|---|---|---|---|---|---|---|---|---|---|---|---|---|---|
| | | IDM | LaDi | Stable | TPD | IDM | LaDi | Stable | TPD | IDM | LaDi | Stable | TPD |
| Shar. ↑ | Baseline | 0.493 | 0.521 | 0.425 | 0.457 | 0.384 | 0.505 | 0.477 | 0.328 | 0.267 | 0.276 | 0.336 | 0.316 |
| | Histogram Eq. | 0.614 | 0.591 | 0.413 | 0.448 | 0.364 | 0.679 | 0.689 | 0.322 | 0.424 | 0.424 | 0.437 | 0.317 |
| | CLAHE | 0.605 | 0.652 | 0.502 | 0.631 | 0.517 | 0.667 | 0.588 | 0.387 | 0.343 | 0.343 | 0.353 | 0.377 |
| | Unsharp Masking | **1.015** | **1.095** | **0.743** | **1.045** | **0.882** | **1.040** | **0.828** | **0.598** | **0.558** | **0.558** | **0.564** | **0.663** |
| | TTA | 0.773 | 0.542 | 0.427 | 0.463 | 0.398 | 0.511 | 0.486 | 0.342 | 0.276 | 0.462 | 0.335 | 0.368 |
| Dist. ↓ | Baseline | 0.045 | 0.043 | **0.039** | 0.044 | 0.055 | 0.051 | 0.043 | 0.062 | 0.063 | 0.063 | 0.041 | 0.047 |
| | Histogram Eq. | 0.048 | 0.049 | 0.046 | 0.049 | **0.032** | **0.027** | 0.047 | **0.036** | **0.032** | 0.062 | 0.047 | **0.033** |
| | CLAHE | 0.075 | 0.085 | 0.062 | 0.081 | 0.087 | 0.089 | 0.083 | 0.102 | 0.093 | 0.093 | 0.086 | 0.095 |
| | Unsharp Masking | 0.053 | 0.057 | 0.050 | 0.056 | 0.066 | 0.060 | 0.052 | 0.073 | 0.067 | 0.067 | 0.055 | 0.061 |
| | TTA | **0.033** | **0.045** | **0.039** | **0.042** | 0.053 | 0.049 | **0.043** | 0.054 | 0.060 | **0.048** | **0.035** | 0.040 |
| TP ↑ | Baseline | 0.716 | **0.735** | 0.763 | 0.760 | 0.634 | 0.686 | 0.676 | 0.668 | 0.723 | 0.723 | 0.776 | 0.753 |
| | Histogram Eq. | **0.746** | **0.735** | 0.773 | 0.752 | **0.656** | 0.686 | 0.691 | 0.675 | 0.739 | **0.769** | 0.780 | 0.742 |
| | CLAHE | 0.709 | 0.723 | 0.757 | **0.753** | 0.635 | 0.695 | 0.698 | 0.667 | 0.718 | 0.718 | **0.765** | 0.741 |
| | Unsharp Masking | 0.686 | 0.697 | 0.735 | 0.726 | 0.611 | 0.673 | 0.677 | 0.646 | 0.690 | 0.690 | 0.740 | 0.703 |
| | TTA | 0.721 | 0.734 | **0.784** | 0.759 | 0.645 | **0.689** | **0.692** | **0.699** | **0.741** | 0.754 | 0.785 | **0.759** |
| PC ↑ | Baseline | 0.809 | 0.843 | 0.847 | 0.845 | 0.801 | **0.817** | 0.818 | 0.793 | 0.827 | 0.827 | 0.845 | 0.835 |
| | Histogram Eq. | **0.838** | 0.841 | 0.843 | 0.841 | **0.812** | 0.813 | 0.816 | 0.796 | **0.841** | 0.841 | 0.846 | 0.840 |
| | CLAHE | 0.839 | 0.842 | 0.846 | 0.841 | 0.807 | 0.811 | 0.819 | 0.791 | 0.829 | 0.829 | 0.841 | 0.833 |
| | Unsharp Masking | 0.824 | 0.831 | 0.842 | 0.834 | 0.791 | 0.802 | 0.816 | 0.784 | 0.817 | 0.817 | 0.840 | 0.825 |
| | TTA | 0.833 | **0.844** | **0.858** | **0.864** | 0.805 | 0.811 | **0.818** | **0.800** | 0.839 | 0.840 | 0.846 | **0.856** |
| CLIP ↑ | Baseline | 0.837 | **0.858** | 0.840 | **0.856** | 0.747 | 0.769 | 0.770 | 0.763 | 0.853 | 0.853 | 0.849 | 0.869 |
| | Histogram Eq. | 0.776 | 0.776 | 0.757 | 0.773 | 0.701 | 0.693 | 0.700 | 0.704 | 0.837 | 0.837 | 0.836 | 0.848 |
| | CLAHE | 0.827 | 0.841 | 0.812 | 0.832 | 0.745 | 0.763 | 0.763 | 0.758 | 0.847 | 0.847 | 0.883 | 0.875 |
| | Unsharp Masking | 0.808 | 0.814 | 0.816 | 0.827 | 0.723 | 0.734 | 0.752 | 0.743 | 0.829 | 0.829 | 0.871 | 0.859 |
| | TTA | **0.848** | **0.858** | 0.845 | **0.856** | **0.750** | **0.775** | **0.771** | **0.764** | **0.861** | **0.884** | **0.889** | **0.889** |

Table 15: Comprehensive Quantitative Results comparing Baseline, TTA, and Adaptation Methods across datasets and methods. The adaptation methods are Histogram Equalization, CLAHE, and Unsharp Masking. Best results in each metric-dataset combination are shown in bold.

distribution, improves in most cases with TTA, showing enhanced output fidelity with input data. TTA significantly improves density and coverage for Stable-VITON on the DeepFashion dataset, demonstrating effectiveness in mitigating domain shift. In the case of IDM-VTON, the generation quality improvement correlates with the improvement in try-on quality as well. TTA seems to underperform metrically on VITON-HD and DeepFashion, despite producing visually superior results across all datasets. Notably, FID does not consistently improve, likely due to limitations in the Inception model's feature representations. Other metrics, independent of specific feature spaces, provide a more reliable assessment of distribution alignment.

## G.1 Failure Cases

While TTA helps improve try-on quality in most cases, it can sometimes fail to correct existing issues. For instance, Fig. 8 shows that the guidance may not help improve a few cases where the sleeve type is incorrectly generated (half-sleeve instead of sleeveless in the top row), incorrect stripe placement and density (second row) and incorrect patterns and textures when there are many intricate patterns observed (bottom row). From the visual results in Fig. 10, 11, 12 and 13, it can also be seen that in some cases, the background regions may get distorted. This is a consequence of incorrect garment masks used, which affects the warping and causes regions outside the garment

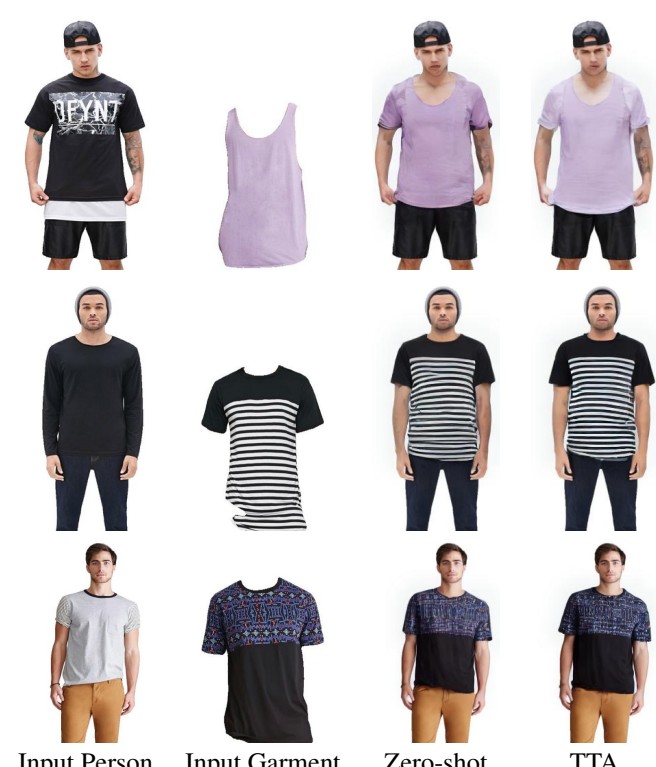

Input Person    Input Garment    Zero-shot    TTA

Figure 8: Visual examples of some failure cases from TPD Yang et al. (2024).

area to get modified by TTA unnecessarily. Using accurate garment masks can help alleviate this issue.

### G.2    ANALYSIS OF VISUAL RESULTS

Figure 10 illustrates examples from the IDM-VTON Choi et al. (2024) model where TTA correctly renders text, ensures consistent stripe spacing, and preserves the garment's color and type to the greatest extent possible. Similarly, Figure 11 demonstrates that TTA improves the Stable-VITON Kim et al. (2024) model by generating more accurate colors and correctly positioning logos.

In Figure 12, the TPD Yang et al. (2024) model fails to reconstruct object shapes, such as striped bands, which are corrected by TTA. Additionally, TTA removes unwanted hallucinations, as seen in the bottom row of Figure 12. Finally, the second-row example in Figure 13 shows TTA accurately rendering garment types, such as a half-sleeve shirt in this case. However, we observe distortions outside the garment region in some cases. This occurs because the baseline model processes areas slightly beyond the exact garment boundaries, as the coarse mask may cover additional regions. In some instances, background color alterations result from the model's warping process. We focus primarily on the garment region since background distortions can be easily removed through postprocessing.

Overall, TTA mitigates various issues in diffusion-based try-on models without requiring domain knowledge of test data, offering a practical plug-and-play solution for enhancing generated image quality.

## H    COMPARISON WITH OTHER ADAPTATION BASELINES

We present a comprehensive comparative analysis of our TTA method against four adaptation baselines: histogram equalization, CLAHE (Contrast Limited Adaptive Histogram Equalization), un-

sharp masking, and the original baseline across all model-dataset combinations. Table 15 summarizes the quantitative results across five key metrics.

## H.1 COMPARATIVE PERFORMANCE ANALYSIS

**Semantic Consistency:** Our TTA method demonstrates exceptional performance in preserving semantic alignment between input garments and generated outputs. TTA achieves the best CLIP consistency scores in all 12 model-dataset configurations, with particularly strong gains on challenging datasets. On DeepFashion, TTA improves CLIP scores by 0.8-2.0% over the strongest baselines, while maintaining consistent improvements across DressCode (0.1-0.6% gains), indicating the robustness of our semantic preservation approach across diverse domain conditions.

**Other Metrics:** Unlike post-processing methods that excel in specific aspects, TTA provides balanced improvements across multiple quality dimensions. While unsharp masking dominates sharpness metrics by design (achieving 1.5-3.8× improvements), TTA delivers competitive sharpness gains (7.74% average improvement) while simultaneously enhancing semantic consistency and geometric accuracy. This balanced performance profile makes TTA particularly valuable for deployment scenarios requiring overall quality enhancement rather than single-metric optimization.

**Geometric Distortion Reduction:** TTA demonstrates superior geometric consistency, achieving the lowest distortion scores in 7 out of 12 configurations. The method shows particular strength on VITON-HD (3/4 wins) and DeepFashion (3/4 wins), reducing distortion by up to 26.7% compared to baselines. On VITON-HD, TTA consistently outperforms all adaptation methods for IDM-VTON (-26.7%), TPD (-4.5%), and ties with baseline performance for Stable-VTON. This indicates that our statistical guidance effectively maintains spatial coherence during the adaptation.

**Dataset-specific Performance:** Performance varies across datasets, revealing important insights about domain characteristics. On VITON-HD (mostly studio conditions), TTA excels in distortion reduction and CLIP consistency, suggesting effective adaptation to in-domain variations. DressCode presents the most challenging scenario, where histogram equalization frequently outperforms other methods in distortion metrics (3/4 wins), indicating that global color adjustments are particularly beneficial for this dataset's diverse poses and garment types. DeepFashion results demonstrate TTA's robustness to real-world conditions, with consistent CLIP improvements and competitive performance across other metrics.

## H.2 METHOD-SPECIFIC ANALYSIS

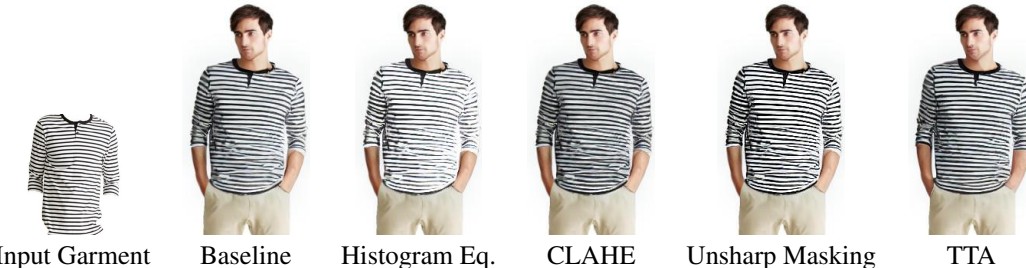

| Input Garment | Baseline | Histogram Eq. | CLAHE | Unsharp Masking | TTA |

Figure 9: Visual results of generations from multiple post-processing-based adaptation baselines. TTA achieves consistent stripe generation, whereas the other methods just vary the contrast or the overall color distribution, offering no visual improvement.

**Post-Processing Method Limitations:** While unsharp masking achieves superior sharpness across all configurations, it shows limitations in semantic preservation and geometric consistency. Despite dramatic sharpness improvements (0.598-1.095 range), unsharp masking fails to achieve the best performance in any other metric, highlighting the trade-off between edge enhancement and overall image quality. Similarly, CLAHE provides reasonable improvements in texture preservation

for certain configurations but generally underperforms in distortion and CLIP consistency, suggesting that local contrast enhancement alone is insufficient for comprehensive quality improvement.

**Histogram Equalization:** Histogram equalization emerges as a surprisingly strong baseline, particularly for distortion reduction on DressCode and DeepFashion. This method achieves the best distortion scores in 6 out of 12 configurations, with notable performance on challenging datasets (DressCode: 3/4 wins, DeepFashion: 2/4 wins). However, histogram equalization consistently underperforms in CLIP consistency, indicating that while it improves geometric accuracy, it may compromise semantic fidelity—a critical limitation for virtual try-on applications.

### H.3 Model-Specific Adaptation Patterns

**IDM-VTON:** IDM-VTON shows the best responsiveness to TTA guidance, achieving the most dramatic improvements in sharpness on VITON-HD (+56.8%) and consistent CLIP gains across all datasets. This suggests that IDM-VTON's architecture is particularly favorable for our guidance approach, possibly due to its semantic information extraction mechanisms aligning well with our multi-domain loss formulation.

**Stable-VTON:** Stable-VTON demonstrates the most consistent baseline performance but shows modest TTA improvements compared to other models. Interestingly, CLAHE achieves competitive results with Stable-VTON on certain metrics, suggesting that this model's pre-trained diffusion backbone may already incorporate some contrast enhancement capabilities that reduce the marginal benefit of our statistical guidance.

**LaDI-VTON and TPD:** LaDI-VTON and TPD show variable responsiveness to different adaptation methods, with LaDI-VTON benefiting significantly from TTA on challenging datasets (DeepFashion CLIP: +3.6%) while TPD shows more consistent but modest improvements. This variability suggests that model architecture characteristics influence adaptation effectiveness, warranting future investigation into architecture-specific guidance strategies.

### H.4 Practical Deployment Implications

**Quality-Efficiency Trade-offs:** Our analysis reveals important trade-offs for practical deployment. While unsharp masking provides superior sharpness with minimal computational overhead, TTA offers comprehensive quality enhancement at the cost of increased inference time (15-25% overhead). For applications prioritizing semantic accuracy and balanced quality improvement, TTA represents the optimal choice. Conversely, scenarios requiring only edge enhancement might benefit from simpler post-processing approaches.

**Domain-Aware Method Selection:** The dataset-specific performance patterns suggest that adaptation method selection should consider target domain characteristics. For controlled studio conditions (similar to VITON-HD), TTA provides optimal performance across most metrics. For diverse real-world scenarios (similar to DeepFashion), TTA maintains its semantic advantages while remaining competitive in other metrics. For applications with highly variable poses and garment types (similar to DressCode), hybrid approaches combining histogram equalization for geometric consistency and TTA for semantic preservation may yield optimal results.

### H.5 Qualitative vs. Quantitative Assessment

**Visual Quality:** While quantitative metrics provide valuable comparative insights, visual inspection reveals a critical limitation of post-processing baselines that metrics alone cannot capture. Despite competitive numerical scores, CLAHE and unsharp masking introduce significant visual artifacts and detail loss that compromise practical applicability. CLAHE, while achieving reasonable texture preservation scores, produces over-enhanced contrast that results in unnatural appearance and loss of subtle garment details. Similarly, unsharp masking, despite superior sharpness metrics, introduces edge artifacts and amplifies noise, particularly in textured regions, leading to visually degraded outputs that would be unacceptable in commercial applications.

**Detail Preservation Analysis:** Our TTA method uniquely preserves fine-grained visual details while achieving metric improvements. Unlike post-processing approaches that operate uniformly across image regions, TTA's statistical guidance respects the underlying garment structure and texture characteristics through distribution-level matching. This results in visually coherent enhancements that maintain garment authenticity.

**Metrics vs. Perceptual Quality Gap:** This observation highlights a fundamental limitation of existing evaluation approaches in virtual try-on: the gap between quantitative metrics and perceptual quality. While unsharp masking achieves superior sharpness scores through aggressive edge enhancement, the resulting images exhibit unnatural over-sharpening that degrades overall visual appeal. Conversely, TTA's modest metric improvements correspond to meaningful perceptual enhancements that preserve natural appearance while addressing domain shift artifacts. This suggests that future evaluation frameworks should incorporate perceptual quality assessment alongside traditional metrics to better capture the requirements of real-world settings.

## H.6 ROBUSTNESS

The comprehensive evaluation across 12 model-dataset combinations and multiple samples per configuration provides strong evidence for TTA's effectiveness. The consistent CLIP performance improvements (all configurations) indicate that our method's semantic preservation capabilities are robust across diverse domain conditions and model architectures. The balanced performance profile across multiple metrics, combined with theoretical convergence guarantees, establishes TTA as a reliable solution for enhancing virtual try-on quality in production environments. Importantly, the qualitative assessment reveals that TTA is the only adaptation method that achieves metric improvements while preserving visual authenticity. This combination of quantitative gains and perceptual quality preservation makes TTA uniquely suitable for commercial virtual try-on applications, where both measurable performance and visual appeal are essential for user acceptance. These results validate our hypothesis that parameter-free statistical guidance can effectively address domain shift challenges in virtual try-on while maintaining computational efficiency and broad applicability across diffusion-based architectures.

| Input Person | Input Garment | Zero-shot output | TTA output |

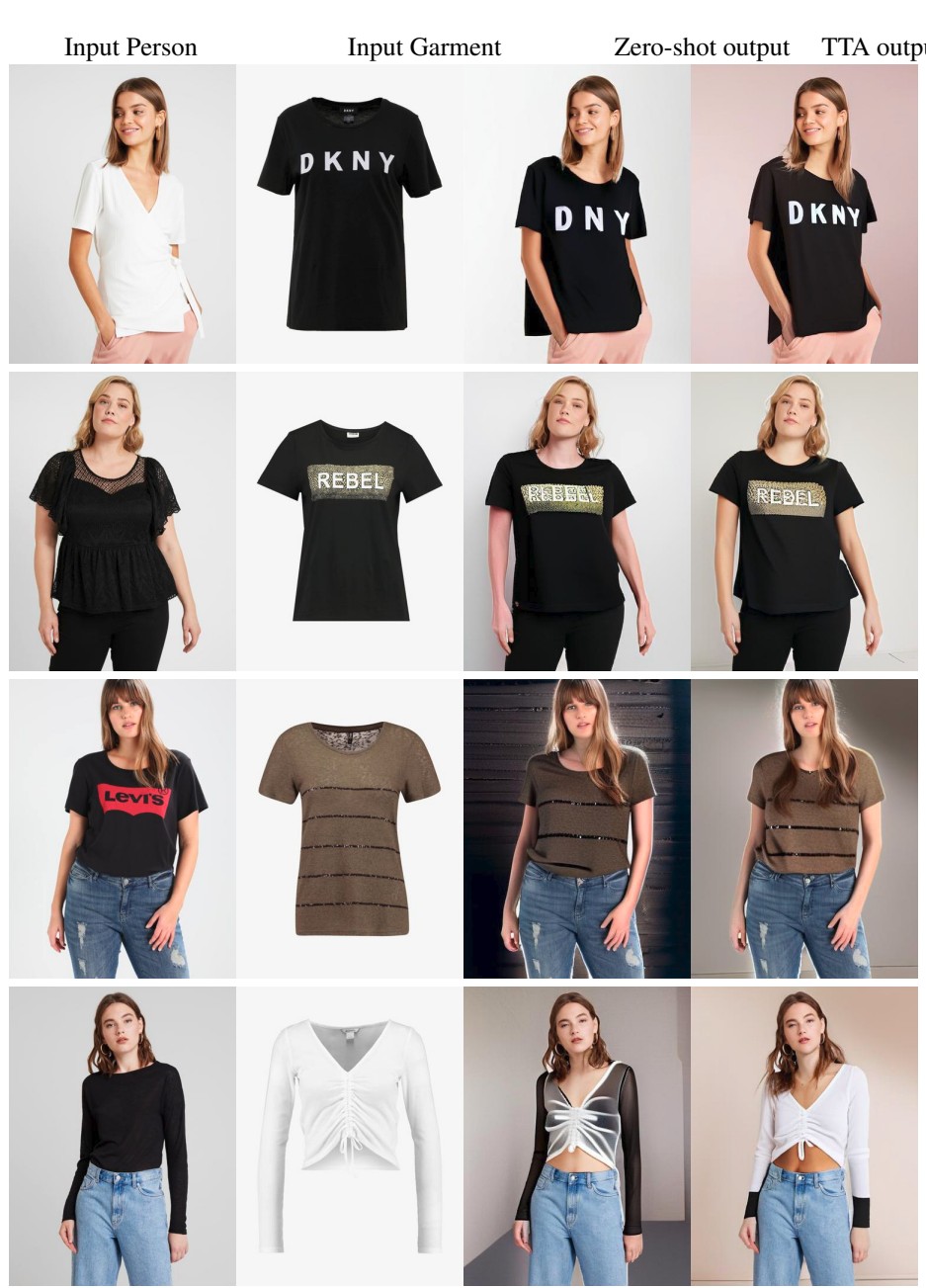

Figure 10: Visual results of the IDM-VTON model along with the corresponding TTA outputs. The text is more accurately generated in the examples in the top-two rows. In the third row from the top, the stripes are consistently placed by TTA and the color is more accurate. In the bottom row, the garment is correctly draped and the colors are consistent.

Input Person       Input Garment       Zero-shot output    TTA output

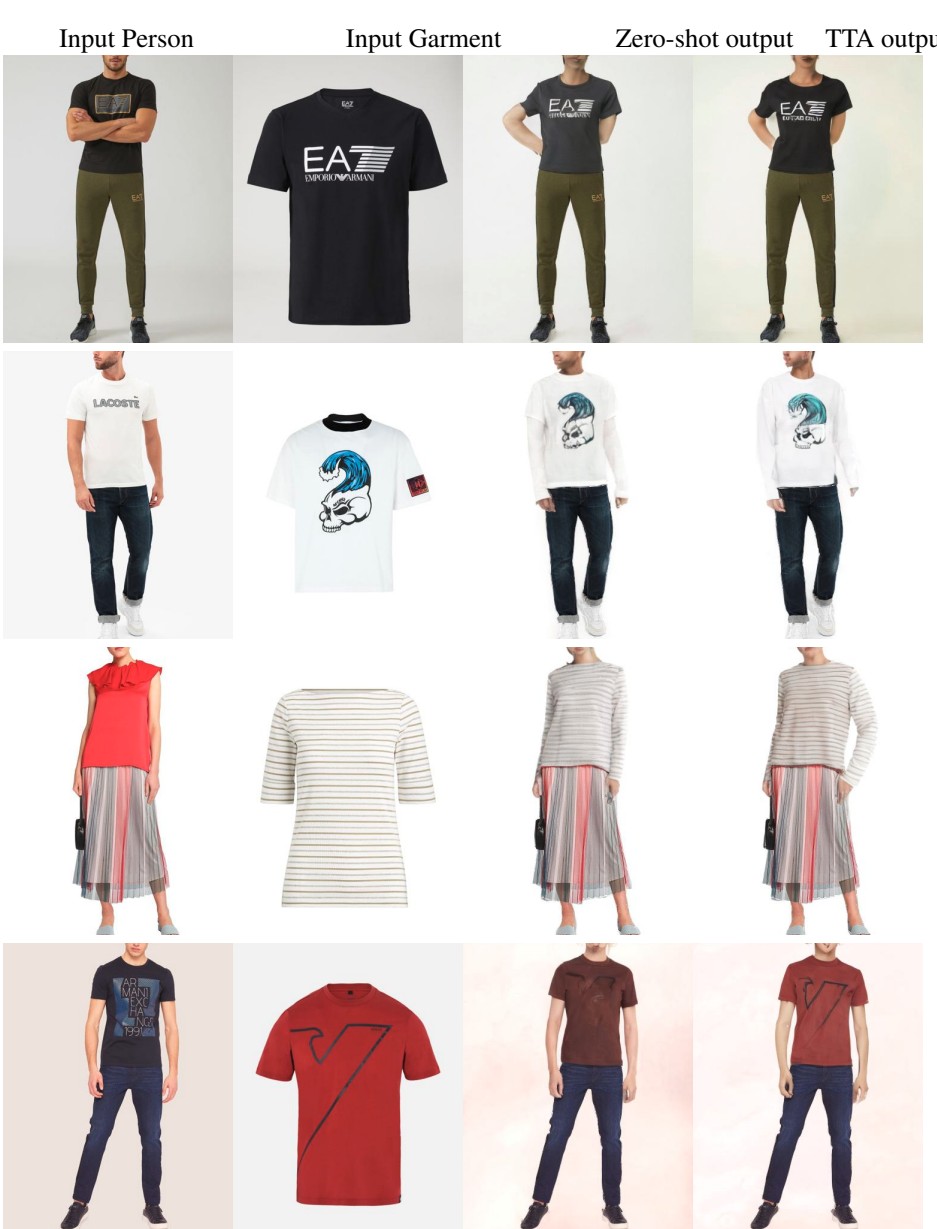

Figure 11: Visual results of the Stable VITON model along with the corresponding TTA outputs. In the top two rows of examples, the color of the logos are more accurately rendered. In the third row from top, the stripes are more consistent in the TTA output with the input garment. In the bottom row, the color is more accurate in the TTA generation, along with better reconstruction of the pattern on the t-shirt.

Input Person    Input Garment    Zero-shot output    TTA output

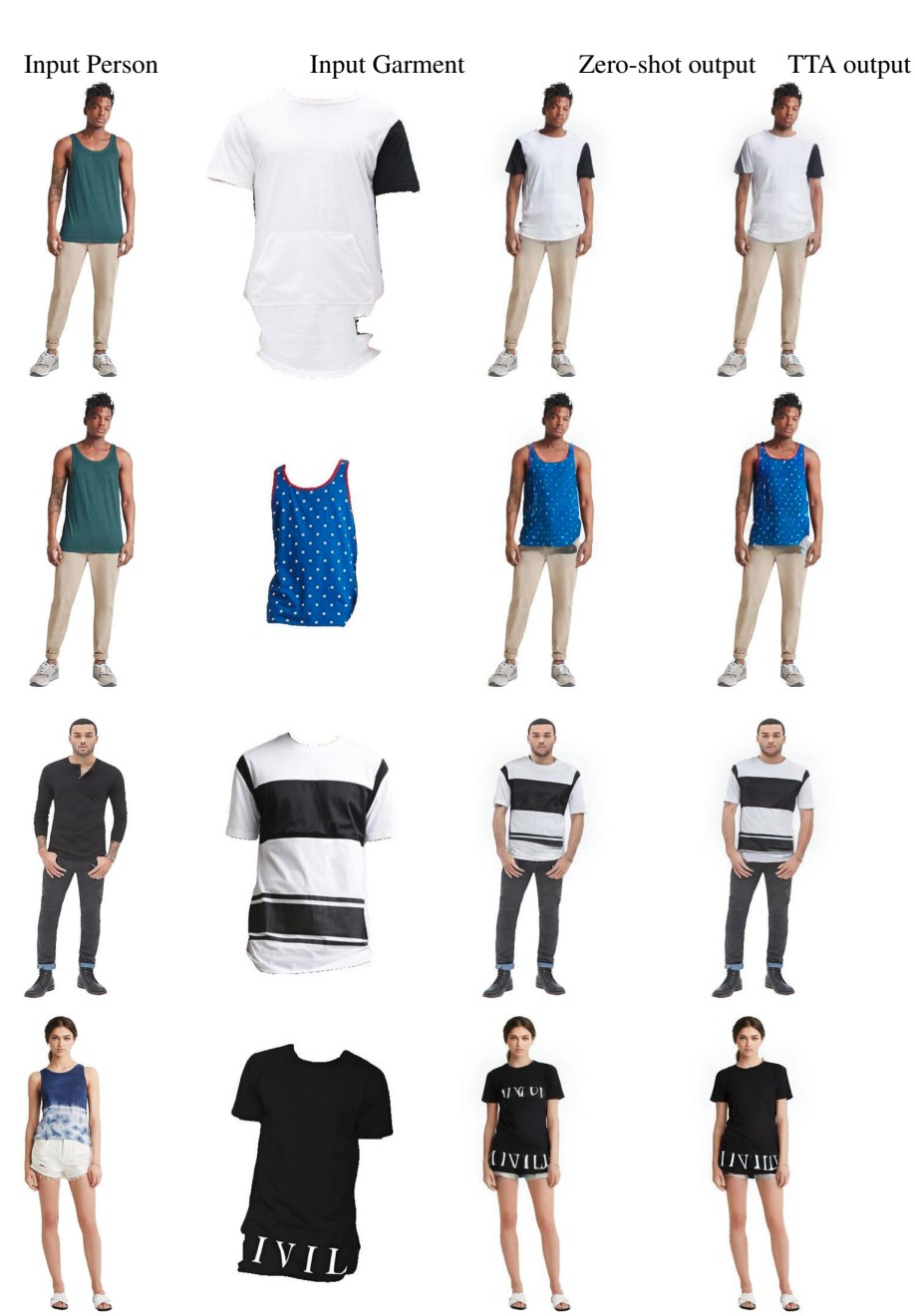

Figure 12: Visual results of the TPD model along with the corresponding TTA outputs. In the top row, the unwanted black color in the right sleeve is reduced in the TTA output. In the second row from top, the white spots are more prominent. In the third row from top, the black striped band is correctly generated with TTA. In the bottom row, the unwanted hallucinated text is removed.

Input Person    Input Garment    Zero-shot output    TTA output

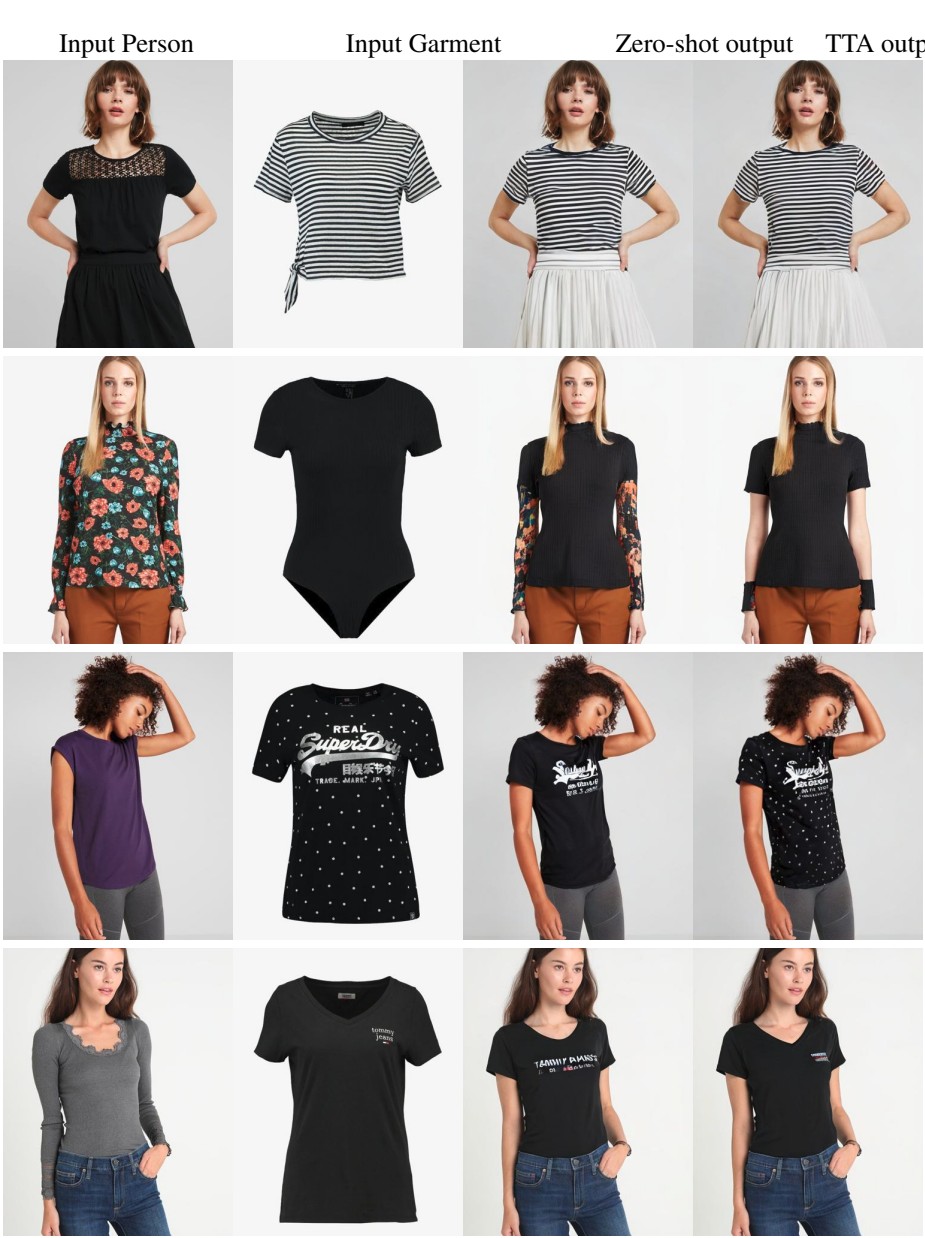

Figure 13: Visual results of the LaDI-VTON model along with the corresponding TTA outputs. In the top row, the stripes have better structure in the TTA output compared to the zero-shot generation. In the second row, the unwanted long sleeve portion from the input garment is effectively removed. In the third row from top, the spots are more prominent, whereas zero-shot fails to produce the spots. In the bottom row, the logo is correctly positioned and its size is accurate as well.

