# OpenReview forum: "Source-Free Test-Time Adaptation for Diffusion-based Virtual Try-On"
_ICLR.cc/2026/Conference — ICLR 2026 Conference Withdrawn Submission_

### Official Review · Reviewer_5Usg · 2025-10-23

**Soundness:** 2
**Presentation:** 2
**Contribution:** 2
**Rating:** 2
**Confidence:** 5

**Summary:**

This paper proposes a novel test-time adaptation (TTA) framework for diffusion-based virtual try-on methods. The approach leverages GDA techniques to perform TTA with a customized loss designed to preserve garment details. In addition, the authors introduce five evaluation metrics for unpaired data evaluation. They claim that the proposed method further improves the performance of prior diffusion-based virtual try-on models.

**Strengths:**

* The paper is easy to follow.
* The idea of applying GDA-based techniques for test-time adaptation is interesting.

**Weaknesses:**

* The stated motivation, to address domain shift, seems misaligned with the actual experiments. The experiments mainly focus on preserving garment details rather than handling clear domain shifts. This discrepancy makes the motivation less convincing.
* The quality of the visualizations is quite poor compared to prior works, making it difficult to assess the claimed improvements. For example, in Figure 4, the proposed TTA appears to degrade visual quality: subfigure (a) shows noticeable artifacts along the boundaries, (b) and (d) generate unwanted hair regions, and (c) exhibits distorted necklines and missing collars. Similarly, the differences in Figure 5 are too subtle to substantiate the claimed gains. While I do not encourage cherry-picked results, the visualizations should at least support the quantitative claims.
* If the main advantage of the proposed approach lies in preserving garment details, prior works such as [a] and [b] have already demonstrated strong performance in this aspect. A more direct comparison with these methods would be helpful.
* Since the adaptation is applied only to garment regions, it may introduce sharp artifacts near garment boundaries, as observed in Figure 4. The paper should discuss potential strategies to mitigate this issue.
* Aligning global distributions does not necessarily ensure visually appealing results, as illustrated by the distorted lines in Figure 4(c). This suggests that high-level distribution alignment alone may not capture fine-grained visual details, which are critical for virtual try-on tasks.
* Although it is valuable to propose additional metrics for unpaired data evaluation, conventional metrics like SSIM, LPIPS and FID should still serve as the primary evaluation criteria for fair comparison with existing methods.

[a] CAT-DM: Controllable Accelerated Virtual Try-on with Diffusion Model, CVPR 2024

[b] Texture-Preserving Diffusion Models for High-Fidelity Virtual Try-On, CVPR 2024

**Questions:**

* How does the proposed method preserve small logos? The Image-Space Distribution Loss and Frequency-Domain Distribution Loss mainly focus on aligning global distributions, while the Local-Structure Distribution Loss models local variations using a Gaussian distribution. It is unclear how this design effectively captures small and uncommon patterns such as logos.
* The paper does not specify how the bounding boxes used in the Frequency-Domain Distribution Loss are obtained. For complex poses (e.g., raised arms), simple bounding boxes may include irrelevant regions. Would this affect the performance?

---

### Official Review · Reviewer_1Kk1 · 2025-10-29

**Soundness:** 2
**Presentation:** 2
**Contribution:** 2
**Rating:** 2
**Confidence:** 3

**Summary:**

The authors propose a test-time adaptation method for improving the performance of virtual try-on diffusion models, reducing distortion and increasing sharpness at a modest computational cost.

**Strengths:**

- The method is logically sound and computationally efficient
- A user study is shown (Table 4) that shows users prefer outputs using the proposed test-time adaptation method

**Weaknesses:**

- The gains from the method are most clear in non-standard metrics such as sharpness and distortion reduction, which the authors themselves introduce. In standard metrics like SSIM, LPIPS, and FID, the gains from the method are much less pronounced and the improvement of the method is not clear. Moreover, all the results for standard metrics are relegated to appendix, instead of being in the main paper and being given a full discussion.
- The most compelling evidence for the advantages of the method is the user study (Table 4) showing a 63% preference rate for using vs not using the method. However, as best I can see this table is floating (not referenced) and its contents are not discussed in the paper. An explanation of how this user study was conducted is also missing.
- The authors do not cite or include results with PromptDresser (Kim et al.; https://github.com/rlawjdghek/PromptDresser), a recent work in this area.

**Questions:**

- How was the user study conducted? How many people were part of the study, and how were the choices presented?
- Is it possible to evaluate the method using the standard metrics in the main paper? Is there a clear trend of improvement from the method and what is its magnitude?
- How does the proposed test-time adaptation work with PromptDresser?

---

### Official Review · Reviewer_SCXa · 2025-10-31

**Soundness:** 3
**Presentation:** 3
**Contribution:** 3
**Rating:** 6
**Confidence:** 3

**Summary:**

The paper tackles domain shift in diffusion-based virtual try-on. When a pretrained try-on model faces unseen poses, lighting, or garment styles at deployment, quality drops (color drift, texture blur, geometric distortions). The authors propose a source-free, training-free test-time adaptation (TTA) that plugs into the diffusion denoising loop and does not update model parameters. At each sampling step, they compute three complementary statistical losses between the input garment and the current generated garment. (1) image-space histogram matching for color/illumination, (2) frequency-domain (DCT) histogram matching for textures, and (3) a local neighborhood distribution (Gaussian/KL) loss for fine structures—and use the gradient to nudge the next step’s sample (Algorithm 1). Experiments across four SOTA VTON backbones and three datasets show consistent gains (e.g., average sharpness +7.74%, distortion −0.95%) without retraining. The method includes a simple convergence guarantee under L-smoothness and a step-size bound.

**Strengths:**

The paper is well-motivated and positions a clear contribution: a source-free, training-free TTA plug-in tailored to diffusion image-to-image VTON rather than classifier-style TTA that relies on entropy/BN statistics (Sec. 1). The method is modular and easy to integrate—external guidance applied at every denoising step without updating base weights—making it model-agnostic and deployment-friendly (Sec. 3; Alg. 1). The tri-objective design is intuitive and complementary: image-space histograms stabilize color/illumination, DCT-space histograms preserve garment textures, and a local neighborhood Gaussian/KL term enforces structural coherence; ablations and a simple convergence argument (L-smoothness) support the design choices (Sec. 3.2–3.3). Empirically, the paper evaluates 4 backbones × 3 datasets and contrasts with post-processing baselines (HE/CLAHE/unsharp), showing consistent improvements without retraining, which strengthens the generality and practicality claims (Experiments; Appendix).

**Weaknesses:**

- The pipeline assumes accurate garment masks and is primarily demonstrated for single-garment settings; the authors acknowledge mask-free and multi-garment extensions as future work, which leaves robustness in cluttered real-world inputs partially unaddressed (Sec. 3.2; Conclusion).
- chosen objectives and reported metrics (e.g., color/texture/structure statistics, sharpness, CLIP-based consistency) may under-capture nuanced garment–body interactions; broader perceptual/user studies or calibration-oriented measures could strengthen claims of holistic visual quality (Sec. 3.2; Experiments/Appendix).

**Questions:**

See weaknesses

---

### Official Review · Reviewer_s9wm · 2025-11-01

**Soundness:** 3
**Presentation:** 3
**Contribution:** 2
**Rating:** 4
**Confidence:** 4

**Summary:**

This paper proposes a test-time guidance method that improves the performance of try-on models without retraining or modifying the original network parameters, yielding notable gains， especially in sharpness.

**Strengths:**

1） This paper proposes using the loss between the input garment and the network’s prediction for guidance. It’s a reasonable improvement of using guidance for try-on models.
2） The paper’s structure is clear and easy to follow. The paper avoids convoluted statements and is concise and easy to understand.

**Weaknesses:**

1) Lack of novelty.It’s common to apply guidance using a strong model (the input garment in this work) and a weaker model (the baseline prediction), similar to AutoGuidance.[1].
2)Lack of Comparision. This paper lacks a comparison with other test-time augmentation methods。
3)Computational Concerns.  As the number of sampling steps increases, will it cause a substantial increase in the computational burden?
	[1] Karras T, Aittala M, Kynkäänniemi T, et al. Guiding a diffusion model with a bad version of itself[J]. Advances in Neural Information Processing Systems, 2024, 37: 52996-53021.

**Questions:**

Please refer to weaknesses part.

---

### Note · Authors · 2025-12-11

I have read and agree with the venue's withdrawal policy on behalf of myself and my co-authors.